EMBO
reports

# A malectin-like receptor kinase regulates cell death and pattern-triggered immunity in soybean

Dongmei Wang[1,2,†], Xiangxiu Liang[3,†], Yazhou Bao[3], Suxin Yang[1], Xiong Zhang[3] (iD), Hui Yu[1], Qian Zhang[3], Guangyuan Xu[3], Xianzhong Feng[1,*] (iD) & Daolong Dou[3,4,**] (iD)

## Abstract

Plant cells can sense conserved molecular patterns through pattern recognition receptors (PRRs) and initiate pattern-triggered immunity (PTI). Details of the PTI signaling network are starting to be uncovered in Arabidopsis, but are still poorly understood in other species, including soybean (*Glycine max*). In this study, we perform a forward genetic screen for autoimmunity-related *lesion mimic mutants* (*lmms*) in soybean and identify two allelic mutants, which carry mutations in *Glyma.13G054400*, encoding a malectin-like receptor kinase (RK). The mutants exhibit enhanced resistance to both bacterial and oomycete pathogens, as well as elevated ROS production upon treatment with the bacterial pattern flg22. Overexpression of *GmLMM1* gene in *Nicotiana benthamiana* severely suppresses flg22-triggered ROS production and oomycete pattern XEG1-induced cell death. We further show that GmLMM1 interacts with the flg22 receptor FLS2 and its co-receptor BAK1 to negatively regulate flg22-induced complex formation between them. Our study identifies an important component in PTI regulation and reveals that GmLMM1 acts as a molecular switch to control an appropriate immune activation, which may also be adapted to other PRR-mediated immune signaling in soybean.

**Keywords** cell death; *Glycine max*; lesion mimic mutant; malectin-like receptor kinase; pattern-triggered immunity

**Subject Categories** Immunology; Microbiology, Virology & Host Pathogen Interaction; Signal Transduction

## Introduction

Soybean is one of the major sources of oil and plant proteins worldwide. Its demand in food, feed, and industrial production has increased stably along with the rapid expansion of the world's population. Soybean diseases, including bacterial blight, phytophthora root rot, and soybean rust, continuously cause great losses to soybean yield and quality worldwide. Traditional disease control mainly relies on chemical and breeding methods, which are sometimes outpaced by the evolution of pathogens. Thus, it is important to study the soybean immune system and to understand how soybean defends itself against pathogens (Whitham *et al*, 2016). Plants are equipped with two layers of immune perception systems: pattern-triggered immunity (PTI) and effector-triggered immunity (ETI; Jones & Dangl, 2006). Plant plasma membrane-localized pattern recognition receptors (PRRs) can sense the presence of pathogens through recognition of microbe-associated molecule, such as bacterial flg22 and *Phytophthora sojae* XEG1, to activate downstream PTI signaling (Chinchilla *et al*, 2006; Wang *et al*, 2018). Successful pathogens can evade plant resistance by secreting effector proteins to suppress plant PTI signaling. Plant intracellular nucleotide-binding and leucine-rich repeat receptors (NLRs) can recognize the presence of microbial effectors to trigger a much stronger ETI response, which is usually accompanied by hypersensitive response (HR; Dou & Zhou, 2012; Jones *et al*, 2016). Both layers of the plant immune system have been extensively studied in Arabidopsis and some crops, such as rice. However, the limited information on soybean immunity focuses mainly on ETI, few on PTI. For example, several NLRs have been cloned and characterized during the last decade (Whitham *et al*, 2016), including Rpg1b/Rpg1r that recognizes AvrB/AvrRpm1 from *Pseudomonas syringae* pv. *glycinea* (*Psg*) (Ashfield *et al*, 2014).

Reports of immune responses mediated by PRRs in soybean are scarce (Whitham *et al*, 2016). Although PTI is not as strong as ETI, it confers a much broader and moderate resistance to most microbes, with potentially lower growth and yield penalty. In plants, PRRs consist of receptor kinase (RK) and receptor protein (RP). RK contains a variable ectodomain potentially involved in ligand perception, a single transmembrane domain, and a cytoplasmic kinase domain for signal transduction. RP contains an

1   Key Laboratory of Soybean Molecular Design Breeding, Northeast Institute of Geography and Agroecology, The Innovative Academy of Seed Design, Chinese Academy of Sciences, Changchun, China
2   University of Chinese Academy of Sciences, Beijing, China
3   Key Laboratory of Pest Monitoring and Green Management, MOA and College of Plant Protection, China Agricultural University, Beijing, China
4   College of Plant Protection, Nanjing Agricultural University, Nanjing, China
    *Corresponding author. Tel: +86 0431 85655051; E-mail: fengxianzhong@iga.ac.cn
    **Corresponding author. Tel: +86 025 84396973; E-mail: ddou@cau.edu.cn
    †These authors contributed equally to this work

ectodomain and a transmembrane domain, but lacks the cytoplasmic kinase domain. Plant RKs and RPs are divided into many subgroups according to the nature of their extracellular domain, such as the leucine-rich repeat (LRR) type, lysine motif (LysM) type, and malectin-like domain type (Tang *et al*, 2017). The Arabidopsis LRR-RK FLS2 and EFR can recognize bacterial flagellin (or the epitope flg22) and ET-Tu (or the epitope elf18), respectively (Chinchilla *et al*, 2006; Zipfel *et al*, 2006), and form a complex with the co-receptor BAK1/SERK3, which acts as a co-receptor for several LRR-type PRRs (Chinchilla *et al*, 2007; Heese *et al*, 2007). Fungal cell wall-derived chitin can be recognized by the LysM-RK protein LYK5, which is in complex with CERK1 (Cao *et al*, 2014). The *Nicotiana benthamiana* receptor protein RXEG1 recognizes the *P. sojae*-derived pattern XEG1 and interacts with the co-receptor NbBAK1 and the adapter NbSOBIR1 (Wang *et al*, 2018). Plant PRRs interact with a subset of proteins, including BIK1 (and its homologs), the NADPH oxidase RbohD, heterotrimeric G proteins (XLG2, AGB1, and AGG1/2), and other regulatory components to constitute a large and dynamic immune receptor complex to activate downstream immune responses (Couto & Zipfel, 2016; Yu *et al*, 2017). PTI includes a series of immune responses, including production of reactive oxygen species (ROS), transient influx of calcium, activation of MAP kinase and calcium-dependent protein kinase cascades, and transcriptional reprogramming (Couto & Zipfel, 2016).

Malectin-like RKs contain two tandem malectin-like domains in their extracellular region to specifically recognize different signal molecules, including rapid alkalinization factors (RALFs; Franck *et al*, 2018). Since the first member was identified in *Catharanthus roseus* (CrRLK1; Schulze-Muth *et al*, 1996), malectin-like RKs have received increasing attention because of their versatile roles in hormone signaling, growth, morphogenesis, reproduction, and stress responses (Franck *et al*, 2018). *Arabidopsis thaliana* BUPS1 and BUPS2 recognize the preferentially pollen-expressed RALF4 and RALF19 peptides to regulate pollen tube rupture and cell wall integrity by forming a receptor complex with ANX1/2, suggesting dynamic interactions among this protein family (Boisson-Dernier *et al*, 2009; Miyazaki *et al*, 2009; Ge *et al*, 2017). FERONIA (FER) recognizes RALF1 to regulate pollen tube reception and cell elongation (Escobar-Restrepo *et al*, 2007; Guo *et al*, 2009; Kessler *et al*, 2010; Haruta *et al*, 2014). Some members also participate in plant immunity. For example, FER positively regulates flg22- or elf18-mediated PTI and facilitates the complex formation between FLS2 and EFR with BAK1 (Stegmann *et al*, 2017). In contrast, ANX1 negatively regulates PTI by inhibiting the flg22-induced FLS2-BAK1 association (Mang *et al*, 2017). Interestingly, a fungal pathogen, *Fusarium oxysporum*, deploys a RALF-like peptide as a virulence effector to hijack plant immunity signaling by targeting FER (Mang *et al*, 2017).

Investigation of plant lesion mimic mutants (*lmms*) has been a powerful forward genetic approach to unravel plant immunity responses and cell death pathways because they display visible spontaneous cell death phenotypes on leaves, that are often related to autoimmunity (Bruggeman *et al*, 2015; Chakraborty *et al*, 2018; Radojicic *et al*, 2018). Here, we report the isolation and functional characterization of a soybean malectin-like RK, GmLMM1, which negatively regulates resistance to bacterial and oomycete pathogens. GmLMM1 directly couples to the PRR complexes, including PRRs, co-receptors, RbohD, and G proteins from soybean, leading to

attenuation of PTI activation. We took advantage of *N. benthamiana*, a model plant for plant immunology, and revealed that GmLMM1 could inhibit the FLS2-BAK1 association under treatment with flg22 and act as a molecular switch to regulate FLS2-BAK1 interaction and immune activation. Thus, we present a novel soybean PTI regulator and reveal a mechanism by which GmLMM1 maintains disease resistance responses at proper levels.

# Results

## Positional cloning of the *GmLMM1* gene

We generated a mutant population derived from ethyl methane sulfonate (EMS)-treated soybean cultivar "Williams 82" and obtained 16 independent lesion mimic mutants (Fig 1A). We noticed that one mutant exhibited constitutively enhanced resistance to *Pseudomonas syringae* pv. *glycinea* (*Psg*), a causal pathogen of bacterial leaf spot on soybean (Fig 1B), suggesting it is an autoimmune-related mutant. Thus, we selected this for further characterization and named it *Gmlmm1-1* (*G. max lesion mimic mutant 1-1*). The mutant had overall shorter plant height, smaller leaf size, fewer branches, and shorter petioles in comparison to the wild type (WT) (Fig 1A). Spontaneous lesions began to appear on the extended leaves and increased following the growth of each leaf, resulting in the phenotypes developing through the entire growth cycle, and only 3–4 new emerging leaves were eventually normal (Fig 1A).

Next, we performed map-based cloning to identify the *GmLMM1* gene. The *Gmlmm1-1* mutant was back-crossed to the wild type (Williams 82) for four generations to purify the genetic background, and then, the progeny was crossed with another soybean cultivar, "Hedou 12". Two F1 seeds were obtained, and their seedlings showed similar phenotypes to that of the WT. In their F2 progenies, 372 individuals were very similar to that of the WT, and 92 exhibited phenotypes of the mutant. The F2 segregation rate fit the expected 3:1 ratio ($x^2 = 3.28$, $df = 1$, $P = 0.07$; Appendix Table S1), indicating the mutation was mono-recessive inheritance. Furthermore, 170 available INDEL molecular markers covering 20 chromosomes were used for rough mapping (Song *et al*, 2015a). We successfully narrowed down the candidate gene to a 2.59 Mb region on chromosome 13 between MOL3684 (15.06 Mb) and MOL0631 (17.65 Mb). In order to further reduce the scope of the candidates, we designed a set of molecular markers (Appendix Table S2). Eventually, the target gene was positioned in a 131 kb region between markers MOL3754 and MOL3780 (Fig 1C). In this region of the soybean cultivar "Williams 82" (Data ref: Schmutz *et al*, 2010), we found 13 genes (Appendix Table S3), including six tandemly repeated homologous genes that share high sequence similarity with Arabidopsis FER and ANX1 (Fig 1D and Appendix Table S4).

Next, we performed whole genome sequencing of the mutant *Gmlmm1-1* and found that only one gene, *Glyma.13G054400*, was mutated in this region. There is a C to T substitution on its second exon, resulting in an early stop at its 636[th] amino acid (Figs 1C and D, and EV1A). To confirm that the lesion mimic phenotype was indeed caused by the mutation of *Glyma.13G054400*, we aimed to find an independent mutant allele for *Gmlmm1* from the above 16 *Gmlmms* using genomic PCR and sequencing. Fortunately, a second mutant allele was successfully identified and named *Gmlmm1-2*,

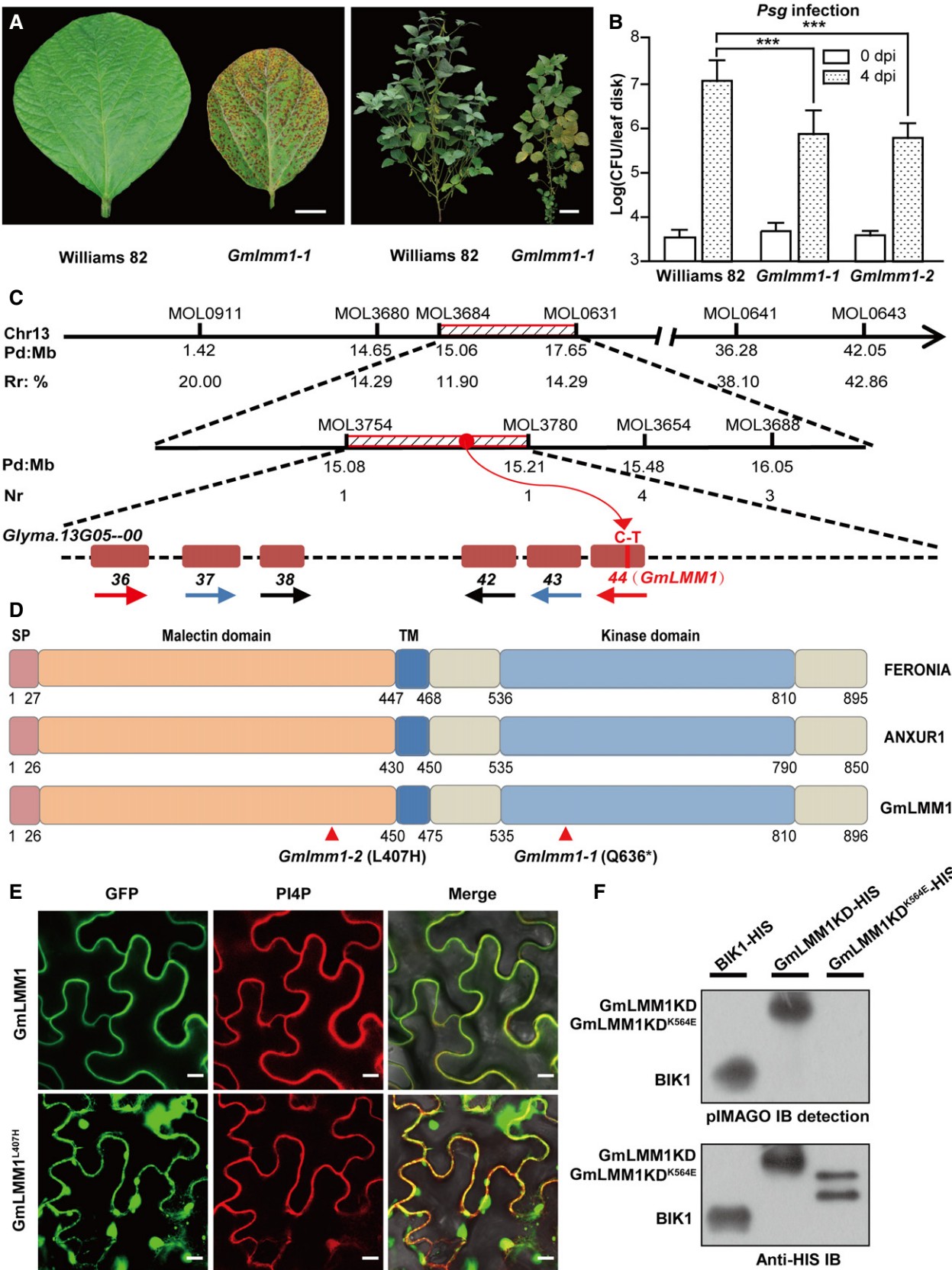

**Figure 1.**

◄

**Figure 1.   Identification and characterization of the *GmLMM1* gene.**

A  Phenotype appearance of the *Gmlmm1-1* mutant. Typical leaves (left panel; scale bar, 1 cm) and whole plants (right panel; scale bar, 10 cm.) were photographed 60 and 118 days after seeding, respectively.

B  Enhanced resistance of *Gmlmm1-1* and *Gmlmm1-2* to *Psg* infection. Leaves from 2-week-old soybean plants were infected with *Psg,* and the bacteria number was determined at 0 and 4 dpi (Mean $\pm$ SD, $n$ = 8, n represents sample number, ***$P$ < 0.001, Student's $t$-test).

C  Physical locations of markers defining the *GmLMM1* region. The chromosomal positions (Pd, physical distance in Mb) of each marker are indicated on chromosome 13 (Chr13) according to the soybean reference genome. Recombination ratio (Rr, %) at the corresponding physical location was calculated using a formula (2 × plant number of genotype HD 12 + plant number of heterozygote genotype)/(2 × total number of plants) for rough mapping. Numbers of recombination (Nr) refers to the number of heterozygous plants at a physical location in the 92 plants for gene mapping. The candidate genes were harbored in a 131 kb region spanned by MOL3754 and MOL3780, in which six homologous genes were tandemly repeated and belonged to the malectin-like gene family in the shadows. The direction of the arrow represents the direction of the gene on the chromosome. Genes with high similarity are indicated by arrows of the same color. The number under the genes corresponds to the gene locus on chromosome 13. The *Glyma.13G054400* (44) gene harbors a mutation (C to T) according to re-sequencing.

D  Comparison of *Arabidopsis* FERONIA, ANXUR1, and GmLMM1. SP and TM indicate the predicted signal peptide and transmembrane domain, respectively. Two corresponding mutated sites in *Gmlmm1-1* and *Gmlmm1-2* are indicated by arrows.

E  Subcellular localization of GmLMM1-GFP and GmLMM1$^{L407H}$-GFP in *N. benthamiana*. PI4P indicates the membrane localization. Scale bar, 10 μm. GFP, green fluorescent protein.

F  Autophosphorylation activity of the GmLMM1 intracellular domain *in vitro*. The proteins purified from *E. coli* were subjected to pMIAGO phosphorylation detection assay (upper panel) and were detected using Western blot (lower panel) with BIK1-HIS as a control. GmLMM1$^{K564E}$ contains a mutation at the predicted ATP binding site.

Data information: The experiments were performed three times (A, B, E) or two times (F), as biological replicates, with similar results.
Source data are available online for this figure.

which harbors a single base (T to A) substitution on its first exon resulting in a leucine to histidine substitution at its 407th amino acid (Figs 1D and EV1A). The *Gmlmm1-2* mutant plants also showed enhanced resistance to *Psg* infection (Fig 1B) and reduced plant height. In addition, the seed number per pod of *Gmlmm1* mutants was significantly reduced. For example, the number of three-seed pods of *Gmlmm1-2* was reduced to half of that of the WT, while the one-seed pod number doubled (Appendix Table S5), indicating that the fertility of sexual organs or early embryo development were also affected in the mutant plants.

We next crossed *Gmlmm1-1* with *Gmlmm1-2* and observed a similar lesion mimic phenotype in the F1 plants (Fig EV1B). Sequencing and restriction enzyme digestion revealed that both mutation sites of the parents were heterozygous in F1 progenies (Fig EV1C). We also analyzed the phenotypes and genotypes of their F2 progenies. All 33 independent F2 plants exhibited similar phenotypes as their parents and contained three different genotypes, $a_1a_1$ (11 lines), $a_1a_2$ (16), and $a_2a_2$ (6), which is consistent with the Mendelian ratio expectation ($a_1a_1$:$a_1a_2$:$a_2a_2$ = 1:2:1. Square test: $x^2$ = 0.76, $df$ = 2, $P$ = 0.69; Fig EV1D and Appendix Table S6). Taken together, we concluded that *Gmlmm1-1* and *Gmlmm1-2* are allelic mutants and therefore named the corresponding gene *GmLMM1*.

### GmLMM1 is a soybean malectin-like RK

Bioinformatic analysis revealed that the *GmLMM1* gene encodes a malectin-like RK, which was first identified as the *Catharanthus roseus* CrRLK1L (Schulze-Muth *et al*, 1996; Franck *et al*, 2018). The closest homologs of GmLMM1 in Arabidopsis are the malectin-like RKs, FER (identity = 50.5%), and ANX1 (identity = 43.8%) (Figs 1D and EV1E). We analyzed the nucleic acid polymorphism in a ~150 kb region from 15.07 Mb to 15.22 Mb on chromosome 13 using 62 varieties of *Glycine soja* and 130 varieties of *G. max*. In the investigated region, the nucleotide diversity (π) of *G. max* and *G. soja* showed no obvious difference, which indicated that

*GmLMM1* is highly conserved in both cultivated and wild soybeans (Fig EV1F). We examined the subcellular localization of GmLMM1 by transiently expressing GFP-tagged GmLMM1 in *N. benthamiana* and observed that GmLMM1-GFP was localized to the plasma membrane (Fig 1E). As the *Gmlmm1-2* mutant harbors a single-site mutation (L407H) in the extracellular region, the localization assay showed that the GmLMM1$^{L407H}$ mutation abnormally aggregates in the cytosol and membrane (Fig 1E). This abnormal subcellular location of GmLMM1$^{L407H}$ may account for its loss-of-function phenotype of *Gmlmm1-2*. The malectin-like RK family proteins contain a C-terminal intracellular kinase domain, which is important for the activation of downstream signaling. An *in vitro* kinase reaction assay revealed that GmLMM1 showed strong autophosphorylation activity. In contrast, a site mutation of GmLMM1$^{K564E}$, which is predicted as the ATP binding site, resulted in lost autophosphorylation activity (Fig 1F). Collectively, we suggest that *GmLMM1* encodes a malectin-like RK with membrane localization and kinase activity.

### Mutations in *GmLMM1* cause cell death mimic phenotypes

To further verify that the mutant phenotype was controlled by a loss-of-function mutation of *GmLMM1*, we designed three CRISPR vectors (C1/2/9) to knock out *GmLMM1* and its neighboring genes with high sequence similarity, *Gm13G053600* and *Gm13G053800* (Fig EV1E and Appendix Table S4). The recombinant plasmids were introduced into wild-type Chinese soybean cultivar, "Dongnong 50 (DN50)", to generate *GmLMM1* mutants. Only one T0 (C1-16) line generated from the C1 vector was obtained. In the T1 lines, C1-16-1 exhibited a lesion mimic phenotype, whereas C1-16-5 was similar to the WT (Fig 2A). Consistent with this, the C1-16-1 line harbored substantial mutations causing a frame shift in the *GmLMM1* gene and a 3-bp deletion in *Gm13G053800*. In contrast, C1-16-5 showed a 3-bp deletion in *GmLMM1* and multiple mutated sites in *Gm13G053800* (Fig 2A). Similarly, the C2 vector targeting *GmLMM1*, *Gm13G053600,* and *Gm13G053800* was introduced into

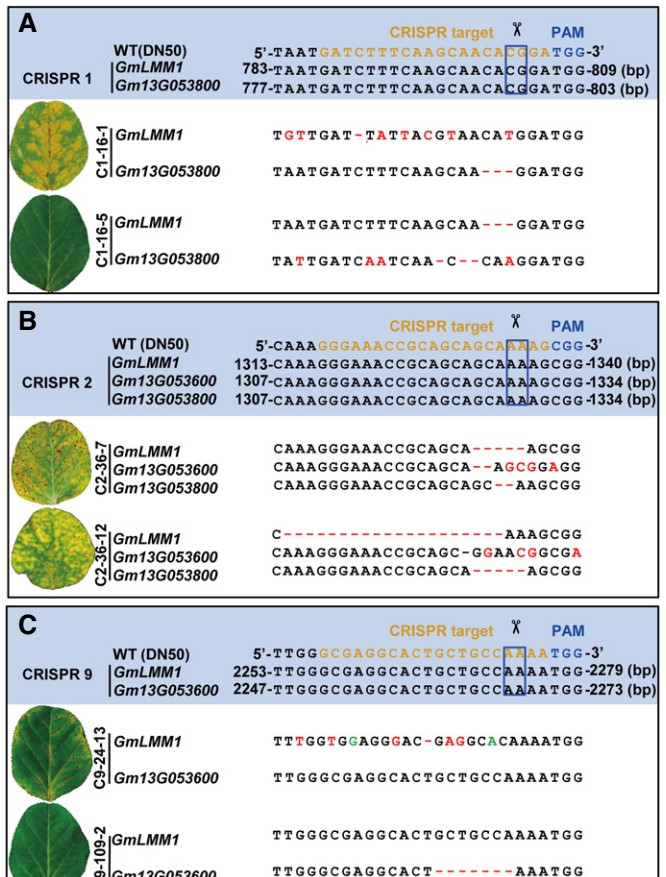

**Figure 2. Multiallelic editing of the *GmLMM1* gene causes the lesion mimic phenotype.**

Three constructs were designed to target *GmLMM1* and its close homologous genes. The CRISPR target sequence is shown in orange, and the corresponding protospacer adjacent motif (PAM) site is shown in blue. The box indicates the putative sites shear cut by cas9. Two representative T1 transgenic lines (aged 1 month) are shown for each construct. The editing results were examined by PCR and sequencing; − means deletion of the corresponding nucleotide. The red or green face-labeled letters indicate the mutated or added nucleotides, respectively.

A  Editing of *GmLMM1* and *Gm13G053800* by construct 1 (C1). The alignment of the sequence and phenotypes of two T1 lines (C1-16-1 and C1-16-5) generated from the same T0 line (C1-16) are shown.

B  Simultaneous edits of *GmLMM1*, *Gm13G053600*, and *Gm13G053800* by the C2 vector.

C  Editing of *GmLMM1* and *Gm13G053600* by C9 vector.

DN50. In the two T1 lines generated, the gene editing events caused a frame shift in all three target genes in the two T1 lines. Consistent with this, both T1 lines exhibited a lesion mimic phenotype (Fig 2B). The C9 vector was designed to simultaneously target *GmLMM1* and *Gm13G053600* and was introduced into DN50 (Fig 2C). The C9-109-2 line harbored a 7-bp deletion in *Gm13G053600*, but *GmLMM1* was not edited. Leaves of the C9-109-2 lines did not show any visible lesion mimic spots (Fig 2C). The C9-24-13 line contained the frame shift mutant of *GmLMM1*, but *Gm13G053600* was not edited. Consistent with this, the C9-24-13 lines showed a visible lesion mimic phenotype (Fig 2C). It is

noteworthy that the lesion mimic phenotype in C9-24-13 plants was not as strong as that in the C1 and C2 CRISPR lines, which could be due to the fact that the C9 vector targets the C terminus of *GmLMM1* (Fig 2C). To further confirm that the lesion mimic phenotype of *Gmlmm1* mutants is caused by autoimmunity-related cell death, we examined cell death in *Gmlmm1-1*, *Gmlmm1-2*, and the CRISPR line C9-24-13. Trypan blue staining showed that all three mutant lines exhibited a stronger cell death phenotype than the corresponding controls (Fig EV2A and B). We further checked the expression of the immune marker genes *GmPR1* and *GmPR2* in the *Gmlmm1* mutants. Both marker genes were highly expressed in *Gmlmm1-1*, *Gmlmm1-2,* and the CRISPR lines (Fig EV2C and D). Taken together, we suggest that the *GmLMM1* gene, but not its two homologous genes, is responsible for the autoimmune-related cell death mimic phenotypes.

## The *Gmlmm1* mutant confers enhanced resistance to bacterial and oomycete pathogens

As the activation of plant immune responses is often accompanied by accumulation of ROS, we investigated ROS accumulation in the *Gmlmm1* mutant lines using DAB staining. Compared to Williams 82, the *Gmlmm1-1* and *Gmlmm1-2* mutants showed significantly enhanced ROS accumulation (Fig 3A). We examined ROS accumulation upon *P. sojae* P7076 (a compatible isolate). The *Gmlmm1-1* and *Gmlmm1-2* mutant lines exhibited significantly enhanced ROS accumulation compared to Williams 82 at 16 hpi (Fig 3B). We next performed RNA-seq analysis on *Gmlmm1-1* and Williams 82 plants. GO term analysis revealed that plant defense-related genes were highly enriched in the *Gmlmm1-1* mutant plants (Fig 3C and Dataset EV1). We then challenged the plants with different pathogens to check the immune responses in the *Gmlmm1* mutant lines and observed that both lines showed greatly enhanced resistance to *Psg* (Fig 1B), as well as to *Pseudomonas syringae* pv. *phaseolicola* (*Psp*) (Fig 3D), which can cause halo blight in bean. Next, we inoculated the *Gmlmm1* mutant lines with *P. sojae* P7076 on leaves and found that the two mutant lines showed much smaller lesions than those on Williams 82 (Fig 3E). Consistent with the lesion size measurement obtained, qPCR analysis revealed that the *Gmlmm1* mutants had lower *P. sojae* biomass than Williams 82 (Fig EV3A). Similarly, the CRISPR line C9-24-13 also exhibited stronger ROS accumulation (Fig EV3B), significantly enhanced resistance to *Psg* (Fig EV3C), and reduced lesion size (Fig EV3D) and *P. sojae* biomass (Fig EV3E) under *P. sojae* infection than DN50. Taken together, the *Gmlmm1* mutants showed enhanced resistance to various pathogens, indicating that *GmLMM1* negatively regulates plant immunity.

## GmLMM1 negatively regulates PTI

Pattern-triggered immunity and ETI are two major layers of plant immune systems. We first examined the well-known bacterial pattern flg22-induced ROS in the *Gmlmm1* mutant lines and found that both mutants showed significantly enhanced ROS production (Fig 4A). The CRISPR line C9-24-13 also showed more ROS accumulation than DN50 upon flg22 treatment (Fig 4B). We also introduced *GmLMM1-HA* into Arabidopsis by *Agrobacterium*-mediated transformation and analyzed flg22-induced ROS production in the

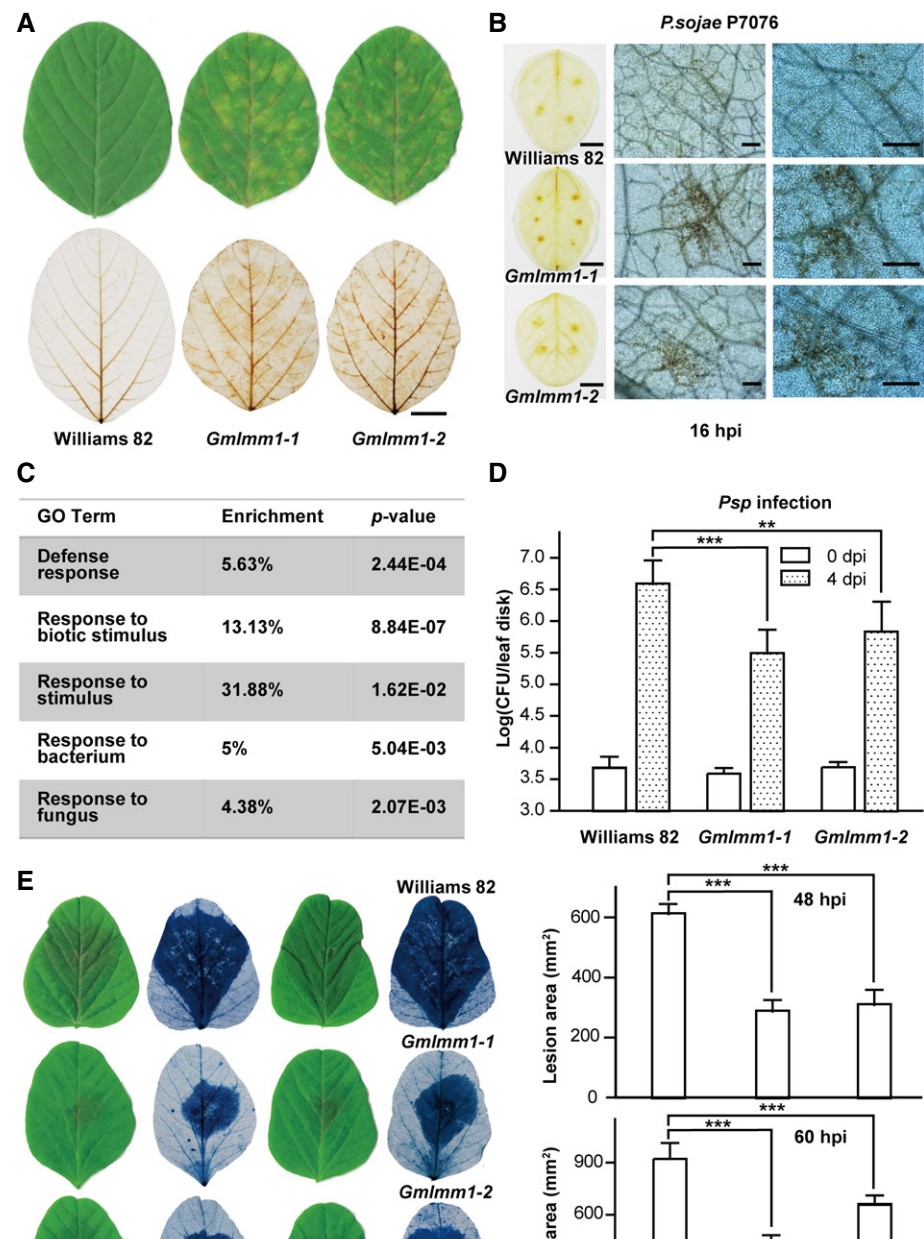

**Figure 3. Gmlmm1 mutant plants exhibit enhanced resistance to bacterial and oomycete pathogens.**

A    ROS accumulation in *Gmlmm1-1* and *Gmlmm1-2* mutant plants. Soybean leaves (aged 3 weeks) were subjected to ROS accumulation examination by DAB staining. Scale bar, 1 cm.

B    ROS accumulation upon *P. sojae* treatment. The leaves (aged 3 weeks) were inoculated with *P. sojae* isolate P7076, and DAB staining was performed at 16 hpi. The left panel was photographed under bright field (scale bar, 1 cm). The middle and right panels were photographed under a microscope (scale bar, 50 μm).

C    Enrichment of defense-related genes in the *Gmlmm1-1* mutant. The *Gmlmm1-1* and WT plants (aged 2 weeks) were subjected to RNA-seq analysis. The genes that were highly expressed in *Gmlmm1-1* compared with the WT (2-fold changes) are presented. The enrichment (%) of the GO terms indicated is presented, and the *P* values are calculated by Fisher's tests. Detailed data are shown in the source data.

D, E    The increased resistance of the mutants to *Psp* (D) and *P. sojae* (E) infection. The indicated plants (aged 2 weeks) were infected with the indicated pathogens (Mean ± SD, $n \geq 8$, n represents sample number, **$P < 0.01$, ***$P < 0.001$, Student's *t*-test). The lesions are shown by trypan blue staining. Scale bar, 1 cm.

Data information: The experiments were performed three times (A, B, D, E), as biological replicates, with similar results. The exact number (*n*) and *P* values are shown in the source data.

Source data are available online for this figure.

transgenic plants. Two independent T2 lines (*GmLMM1*-L3 and *GmLMM1*-L6) with similar GmLMM1-HA expression levels were subject to flg22-induced ROS analysis. We observed that both T2 lines showed significantly reduced flg22-triggered ROS production in Arabidopsis (Fig 4C), suggesting that GmLMM1 can suppress flg22-induced immunity.

Next, we validated the regulation activity of PTI by GmLMM1 using a transient expression assay in *N. benthamiana*. We observed that the expression of GmLMM1 strongly suppressed flg22-induced ROS accumulation (Fig 4D). However, chitin-induced ROS accumulation was suppressed only slightly by GmLMM1 (Fig 4E), indicating that GmLMM1 may participate in different PTI pathways in distinct manners. *P. sojae*-derived XEG1 is another microbial pattern and triggers response in *N. benthamiana* (Wang *et al*, 2018). We observed that GmLMM1 also strongly suppressed XEG1-induced cell death (Fig 4F). The mutant (GmLMM1$^{K564E}$) with loss-of-kinase activity exhibited similar activities as the WT in inhibiting flg22/XEG1-triggered immunity (Fig 4D–F), suggesting that the suppression is not dependent on its kinase activity. In addition, to address whether GmLMM1 can regulate ETI, we co-expressed *P. infestans Avr3a* and potato *R3a* genes that can trigger HR (Bos *et al*, 2009) and observed that GmLMM1 failed to suppress Avr3a/R3a-mediated cell death (Fig EV4A). We then tested *P. syringae* effectors AvrB and AvrRpt2, which can induce HR in *N. benthamiana* and Arabidopsis plants (Grant *et al*, 1995; Mackey *et al*, 2003). As the results shown, AvrB and AvrRpt2-induced cell death was not affected by the expression of GmLMM1 (Fig EV4B and C). Consistent with this, AvrB and AvrRpt2-induced cell death occurred normally in GmLMM1-transgenic Arabidopsis leaves (Fig EV4D). Taken together, these results indicate that GmLMM1 can inhibit the plant PTI responses in a kinase-independent manner.

### GmLMM1 directly couples to the PRR complexes

Next, we sought to identify the GmLMM1-interacting protein by protein immunoprecipitation and mass spectrometry (IP-MS) analysis using the Arabidopsis protoplast assay. In the candidate list, we observed a subset of proteins, including AGB1 (G protein β subunit), RbohH (a homology of RbohD), MSS1/STP13, and five RKs (Fig 5A and Dataset EV2), which were previously reported as parts of the PRR complexes (Kadota *et al*, 2014; Li *et al*, 2014; Liang *et al*, 2016; Yamada *et al*, 2016). We checked whether GmLMM1 interacts with these putative candidates. G proteins and RbohD are well-studied components in the PRR complexes and are key regulators of PTI signaling. We chose the closest homologs of Arabidopsis G proteins and RbohD from soybean (Fig EV5A and B), and performed protein–protein interaction assays. GmLMM1 strongly interacted with the non-canonical XLG protein (GmXLG), GmGα, GmGβ, and GmRbohD. As a negative control, GmILPA1, an APC8-like protein, did not interact with GmLMM1 (Fig EV5C and D). These results indicate that GmLMM1 is likely a component in PRR complexes of soybean.

From the data presented above, we reasoned that GmLMM1 may directly interact with PRRs. Luciferase complementation assays showed that GmLMM1 interacts strongly with GmFLS2 (Gm08G083300, a putative receptor for flg22 from soybean) (Fig 5B), the closest homolog of Arabidopsis FLS2 (Fig EV5E). NbFLS2 and NbRXEG1 are receptors for bacterial flagellin and oomycete XEG1, respectively (Chinchilla *et al*, 2006; Wang *et al*, 2018). Our luciferase complementation assays also indicated that GmLMM1 interacts with both NbFLS2 and NbRXEG1 (Fig 5B). Immunoblot analysis showed the proteins were normally expressed in the above luciferase complementation assays. We further performed Co-IP assays in

*N. benthamiana* by *Agrobacterium*-mediated transient expression to confirm these interactions. The Co-IP assays showed that GmLMM1 associates with GmFLS2, NbFLS2, and NbRXEG1 (Fig 5C and D).

Both flg22- and XEG1-triggered immunity require the co-receptor BAK1 (Chinchilla *et al*, 2007; Heese *et al*, 2007; Wang *et al*, 2018). We then examined whether GmLMM1 also interacts with the co-receptor BAK1. Gm15G051600 and Gm08G180800 are the two closest homologs of Arabidopsis BAK1 (Fig EV5F), which are named GmBAK1a and GmBAK1b, respectively. The luciferase complementation assays showed that GmLMM1 interacted strongly with both of them, as well as with NbBAK1 (Fig 5E). Immunoblot showed the proteins were normally expressed in the luciferase complementation assays. These interactions were further confirmed by Co-IP assays (Fig 5F). Taken together, we conclude that GmLMM1 is a component of PRR complexes.

### GmLMM1 inhibits flg22-induced FLS2-BAK1 interaction in *N. benthamiana*

As GmLMM1 participates in the PRR receptor complexes and negatively regulates PTI in a kinase activity-independent manner, we hypothesized that GmLMM1 might regulate PRR complex formation to attenuate PTI signaling. Using Co-IP assays, we indeed observed that GmLMM1-NbFLS2 interaction was significantly enhanced by flg22 (Fig 6A). In contrast, flg22 did not affect the GmLMM1-NbBAK1 interaction (Fig 6B). We inferred that GmLMM1 could affect flg22-triggered NbFLS2-NbBAK1 interaction. We therefore examined the NbFLS2-NbBAK1 interaction when GmLMM1 or GmLMM1$^{L407H}$ was present. As Fig 6C shows, flg22 triggered a rapid interaction between NbFLS2 and NbBAK1 to initiate flg22-induced downstream immune signaling. The NbFLS2-NbBAK1 interaction was severely suppressed by GmLMM1, but not by GmLMM1$^{L407H}$ (Fig 6C), indicating that GmLMM1 reduces flg22-induced immune responses by inhibiting NbFLS2-NbBAK1 association. We reasoned that GmLMM1 may suppress the GmFLS2-GmBAK1 interaction. To verify this, we co-expressed GmFLS2 and GmBAK1a in *N. benthamiana* plants, treated with flg22, and subjected to Co-IP assays. We observed that flg22 rapidly triggered the GmFLS2-GmBAK1a interaction (Fig 6D), indicating that GmFLS2 responds to flg22 treatment. We noticed that GmLMM1 strongly suppressed the GmFLS2-GmBAK1a interaction upon flg22 treatment (Fig 6D). GmBAK1b is another BAK1 candidate in soybean, which also interacts with GmLMM1 (Fig 5E and F). Thus, we next checked the effect of GmLMM1 on their interaction. As expected, the interaction was also strongly suppressed by GmLMM1 (Fig 6E). Collectively, GmLMM1 can inhibit the complex formation of FLS2 and BAK1. Thus, we reveal a novel regulator from soybean that functions as a switch to control complex formation between PRRs and their co-receptors, thereby finely tuning PTI activation in soybean (Fig 6F).

## Discussion

We identified two soybean lesion mimic mutants and cloned the responsible gene, *GmLMM1*, a member of the malectin-like RK family. In *Gmlmm1-1* plants, a point mutation resulted in a truncated protein without the intracellular domain, whereas a point mutation in *Gmlmm1-2* plants caused a residue change at the

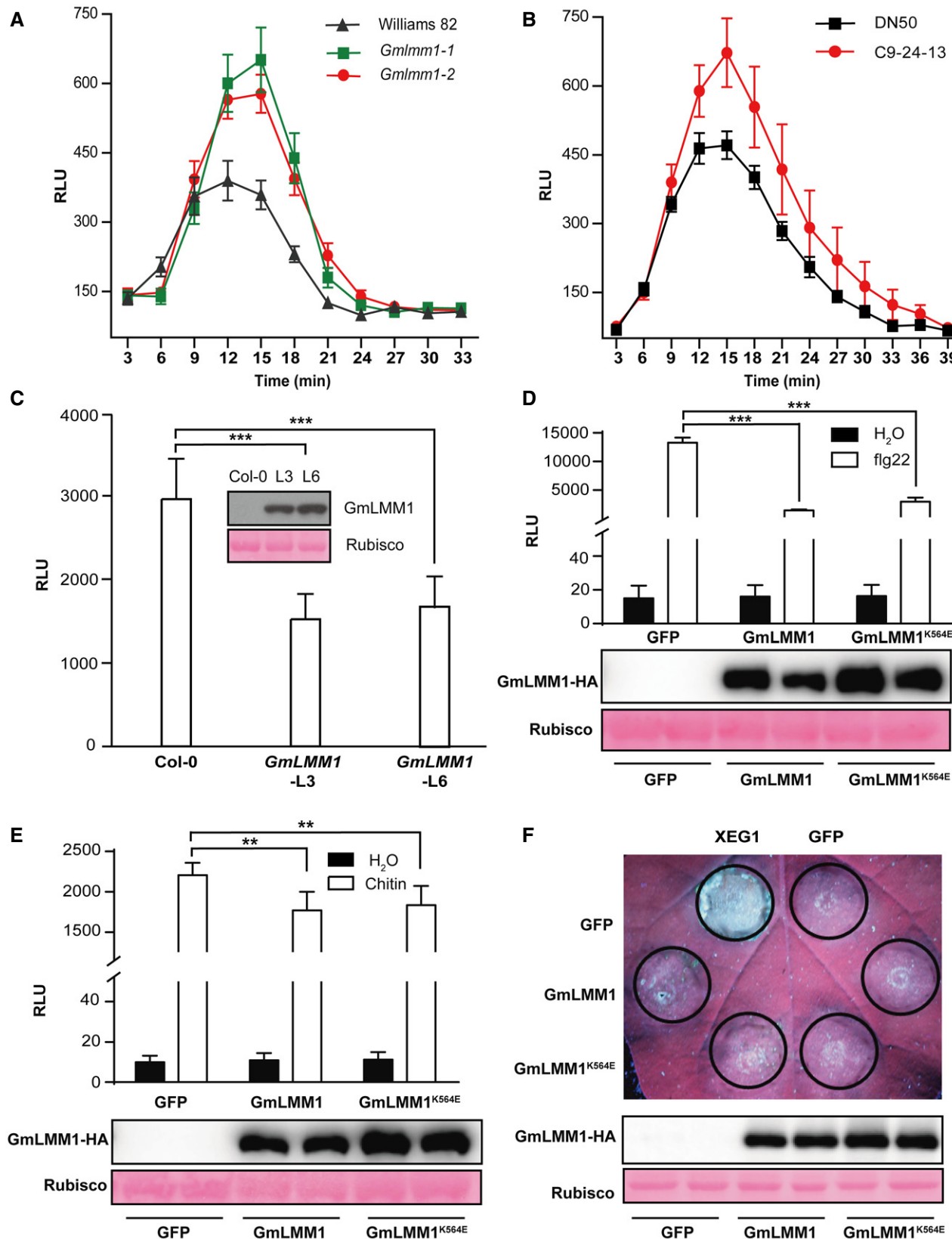

Figure 4.

**Figure 4.  GMLMM1 negatively regulates PTI responses.**

A  Flg22-induced ROS burst in *Gmlmm1-1*, *Gmlmm1-2*, and Williams 82. The indicated leaves (aged 2 weeks) were subjected to flg22-induced ROS examination, and the peak relative luminescence unit (RLU) value was recorded (Mean ± SEM, *n* ≥ 12, n represents sample number).

B  Flg22-induced ROS burst in the CRISPR lines C9-24-13 and DN50. The indicated leaves (aged 2 weeks) were subjected to flg22-induced ROS examination, and the relative RLU value was recorded (Mean ± SEM, *n* ≥ 9, n represents sample number).

C  Suppression of flg22-induced ROS production in *Arabidopsis* by GmLMM1. The *GmLMM1* gene was introduced into *Arabidopsis* WT (Col-0) plants by *Agrobacterium*-mediated transformation. Two independent T2 lines were subjected to flg22-induced ROS examination (Mean ± SD, *n* ≥ 8, n represents sample number, ***P < 0.001, Student's *t*-test). The protein expression is shown.

D, E  Suppression of flg22-induced (D) and chitin-induced (E) ROS production in *N. benthamiana* by GmLMM1. The indicated constructs were transiently expressed by *Agrobacterium*-mediated transient expression for 2 days and subjected to flg22-induced ROS examination. GmLMM1$^{K564E}$ contains a mutation at the predicted ATP binding site without kinase activity. GFP was used as a control (Mean ± SD, *n* = 6, n represents sample number, **P < 0.01, ***P < 0.001, Student's *t*-test). The protein expression is shown in the lower panel.

F  Inhibition of XEG1-induced cell death in *N. benthamiana* by GmLMM1. The indicated constructs were transiently expressed in *N. benthamiana* 1 day before infiltration of *Agrobacterium* containing the *XEG1* or *GFP* vector. The cell death phenotypes were visualized 5 days later. Protein expression of GmLMM1-HA and GmLMM1$^{K564E}$-HA is shown in the lower panel.

Data information: All the experiments were performed three times (biological replicates) with similar results. The exact n, SEM values, and P values are shown in the source data.
Source data are available online for this figure.

juxtamembrane region and altered proper localization. Together with the CRISPR mutant lines obtained, we assumed that *GmLMM1* function is impaired in these mutants, and *Gmlmm1-1* and *Gmlmm1-2* are two independent loss-of-function mutations. Furthermore, we showed that GmLMM1 regulates PTI by association and interference with the formation of PRR complexes. GmLMM1 directly interacts with FLS2 and BAK1 from both soybean and *N. benthamiana* and suppresses flg22-induced immune responses. Consistent with this, the *Gmlmm1* mutant lines showed enhanced flg22-induced ROS burst and were more resistant to bacterial and oomycete pathogens. Overexpression of GmLMM1 in *N. benthamiana* and Arabidopsis plants suppressed flg22- and XEG1-induced PTI responses. Thus, GmLMM1 may control the appropriate activation of PTI responses in soybean.

Plant lesion mimic mutants (*lmm*s) with spontaneous HR-like lesions have emerged as a versatile source for the elucidation of plant immunity responses and a potential donor for disease resistance breeding. To date, several hundreds of *lmm*s have been identified genetically and dozens of corresponding genes have been cloned, mainly in Arabidopsis and rice (Bruggeman *et al*, 2015; Chakraborty *et al*, 2018; Radojicic *et al*, 2018). Most *lmm*s show enhanced disease resistance to various pathogens, which is often accompanied by fast and strong ROS accumulation, HR development, and expression of defense-related genes. The responsive genes are mostly related to chloroplasts activity, metabolism of sphingolipids and fatty acids, cytoplasmic nucleotide-binding leucine-rich repeat domain receptors, and hub regulators of stress signaling pathways. Characterizations of these genes and their close homologs have enabled better understanding of plant disease resistance mechanisms and plant cell death. Take soybean as an example, the reverse genetic approach has revealed that silencing of *GmMPK4* or *GmSSI2* gene in soybean plants develops the similar phenotypes as Arabidopsis *mpk4* or *ssi2* loss-of-function mutants, including spontaneous cell death on the leaves and enhanced resistance to pathogens (Kachroo *et al*, 2008; Liu *et al*, 2011). Here, we identified and characterized *Gmlmm1* mutants, and found that a mutation in a soybean malectin-like RK gene conferred resistance against bacterial and oomycete pathogens using forward genetic tactics. Our results suggest that lesion mimic mutants can be

effectively used to identify additional important genes in soybean immunity responses and cell death.

The RK family is the largest group of receptors in plants and of paramount importance for PTI. To date, few *RK* genes have been identified from plant *lmm*s although many of them are negative regulators of the cell death pathway and autoimmunity. For example, BAK1 is a general co-receptor for multiple LRR-type receptors that have various functions, whereas the *bak1* mutant only develops necrosis upon pathogen infection and the *bak1 bkk1* double mutant exhibits uncontrolled cell death (Kemmerling *et al*, 2007; Roux *et al*, 2011). One exception is the disruption of Arabidopsis BIR1, a BAK1-interacting RK with five extracellular LRR repeats, which may cause similar phenotypes as *Gmlmm1* in this study, including cell death, smaller leaves, shorter hypocotyls, and enhanced disease resistance pathways. BIR1 may participate in preventing the activation of NLR protein-mediated resistance (Gao *et al*, 2009).

We proved that GmLMM1 is involved in maintaining PTI activation through several lines of evidence, including the enhanced immune responses in several soybean mutants, inhibition of flg22-mediated ROS in Arabidopsis and *N. benthamiana*, suppression of XEG1-, but not Avr3a-R3a/AvrB/AvrRpt2-induced cell death, and association with the PRR complexes. Thus, soybean GmLMM1 represents an example of loss-of-function of a plant RK causing the cell death phenotype. Considering that the tandem duplication of GmLMM1 and its five homologs on chromosome 13 in soybean is similar to resistance gene clusters in plant genomes (Deng *et al*, 2017), we speculated that this may facilitate the subfunctionalization or coordinated gene expression for soybean. Although we revealed the roles of GmLMM1 on PTI regulation, we are uncertain of the exact mechanisms about that the spontaneous cell death is regulated by Gmlmm1 through its association with the PRRs or co-receptors. It has been reported that the cell death-related RKs are involved in the PRR complex. For example, expression of several Arabidopsis Cys-rich RKs induces cell death in *N. benthamiana* and some of them also interact with FLS2 and BAK1 (Yadeta *et al*, 2017). The monocot-specific RK SDS2 interacts with the E3 ligase SPL11 and OsRLCK118, and positively regulates plant cell death and immunity in rice (Fan *et al*, 2018). The SERK family proteins

A

| IP-MS identification for GmLMM1 interacting candidate | Mascot score |
|---|---|
| AT5G39000, receptor-like protein kinase At5g39000 | 445 |
| AT2G01210, leucine-rich repeat transmembrane protein kinase, putative | 300 |
| AHA2, ATPase 2, plasma membrane-type | 288 |
| MSS1, Sugar transport protein 13 | 130 |
| AT5G48740, LRR receptor-like serine/threonine-protein kinase At5g48740 | 117 |
| AGB1, Guanine nucleotide-binding protein subunit beta | 70 |
| RbohH, NDHH NAD(P)H-quinone oxidoreductase subunit H | 52 |
| AT3G59350, PTI1-like tyrosine-protein kinase 3 | 52 |
| AT4G04960, L-type lectin-domain containing receptor kinase VII.1 | 32 |

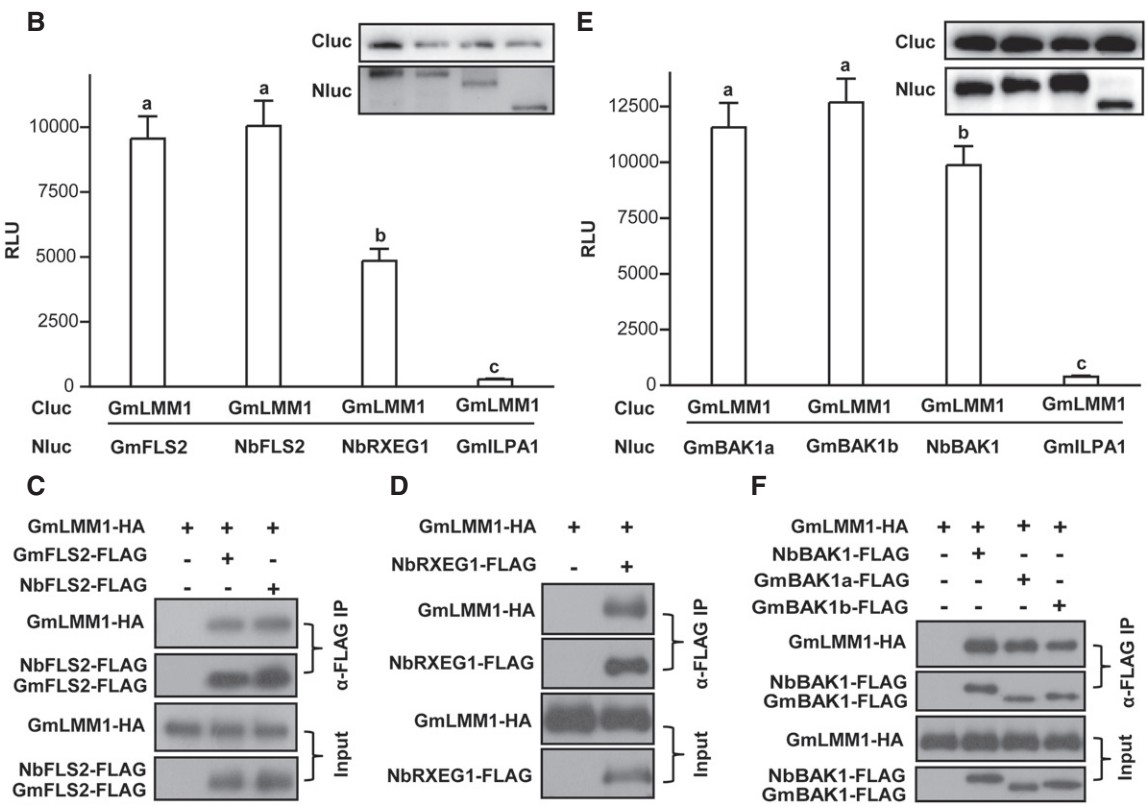

**Figure 5. GMLMM1 is involved in the PRR complexes.**

A   Identification of GmLMM1-interacting proteins by IP-MS analysis. GmLMM1-FLAG was transiently expressed in Arabidopsis protoplasts, purified by anti-FLAG M2 agarose, and subjected to mass spectrometry analysis. The complete list is shown in the source data.

B   Interactions of GmLMM1 with PRR receptors. The indicated constructs were transiently expressed in *N. benthamiana* leaves and subjected to luciferase complementation assay. GmFLS2 (Gm08G083300) is the closest homolog of FLS2 in soybean. GmILPA1 was used as a negative control. The protein interaction intensity is shown by the relative luminescence unit (RLU) (Mean ± SD, $n \geq 6$, n represents sample number, $P < 0.05$, Student's *t*-test, different letters indicate significant difference). Immunoblots show the levels of protein expression.

C   Co-IP of GmLMM1 with NbFLS2 and GmFLS2. The indicated constructs were transiently expressed in *N. benthamiana* by *Agrobacterium*-mediated transient expression for 2 days and the protein interactions were examined by Co-IP assay.

D   GmLMM1 interacts with NbRXEG1 in *N. benthamiana* plants.

E   Interaction of GmLMM1 with NbBAK1, GmBAK1a, and GmBAK1b. The experiment and statistical testing were performed as in (B). Immunoblots show the levels of protein expression.

F   GmLMM1 interacts with NbBAK1, GmBAK1a, and GmBAK1b. The indicated constructs were transiently expressed in *N. benthamiana*, and the interactions were examined by Co-IP assays.

Data information: The experiments were performed three times (B–E) or two times (F), as biological replicates, with similar results.
Source data are available online for this figure.

including BAK1 are reported to function in regulation of cell death processes (Chinchilla *et al*, 2009). We speculate that SERKs may be involved in the above RK-mediated cell death. It will be interesting to examine whether soybean cell death-related components, e.g., BAK1 and other SERK family proteins, participate in cell death control through coordination with GmLMM1.

The malectin-like RK family was reported to recognize a family of secreted peptides named RALFs to regulate root and pollen tube growth and immunity (Boisson-Dernier *et al*, 2009; Miyazaki *et al*, 2009; Ge *et al*, 2017; Mang *et al*, 2017; Stegmann *et al*, 2017). Arabidopsis FER and ANX1 share high sequence similarity and can directly associate with the FLS2-BAK1 receptor complex but regulate flg22-induced PTI in a mechanistically antagonistic manner (Mang *et al*, 2017; Stegmann *et al*, 2017; Xiao *et al*, 2019). FER enhances FLS2-BAK1 interaction to boost PTI (Stegmann *et al*, 2017), while ANX1 inhibits this association, thus negatively regulating PTI (Mang *et al*, 2017). Intriguingly, GmLMM1, sharing close sequence and functional similarity to Arabidopsis ANX1 and FER, forms a PRR complex with FLS2-BAK1 to regulate PTI in soybean, suggesting GmLMM1 functions are evolutionarily conserved across plant species. However, GmLMM1 is distinct from FER and ANX1 in several aspects. For example, mutants of GmLMM1 in two independent alleles and several CRISPR lines of soybean led to an obvious HR-mimic phenotype, which was not found in the mutants of *fer*, *anx1*, or even *anx1 anx2* double mutant. Considering that soybean exhibits high gene expansion of malectin-like RKs, we speculate that GmLMM1 is an essential regulator to maintain PTI levels in soybean, which could be supported by the fact that it is highly conserved in all the soybean cultivars with genome sequences, including wild soybean (*G. soja*). Thus, the gene will be a desirable target for future molecular breeding of disease resistance. Mechanistically, association of GmLMM1 with BAK1 is constitutive and is not affected by flg22 treatment, in contrast to flg22-induced FER/ANX1 interaction with BAK1 (Mang *et al*, 2017; Stegmann *et al*, 2017). GmLMM1 inhibits flg22- and XEG1-mediated immune responses, but only weakly suppresses chitin-induced immunity. We suggest that GmLMM1 could regulate PTI when pathogens are absent or vanquished to balance the tradeoff between growth and resistance. These results indicate that the malectin-like RKs in different organisms show both common and specific roles in immune signaling.

We present that interaction of BAK1 with GmLMM1 is constitutive, which is analogous to what was previously reported for IMPAIRED OOMYCETE SUSCEPTIBILITY1 (IOS1). IOS1 contains a malectin-like domain followed by two LRR motifs in the ectodomain and positively controls FLS2-BAK1 complex formation to regulate PTI (Yeh *et al*, 2016). As the BAK1-FLS2 association is elicited by flg22 treatment, we hypothesize that membrane-localized GmLMM1 may govern PTI activation by regulating early PRR complex formation events or relieving PRR complex formation when pathogens are vanquished by plant immunity. The flg22-mediated interaction of FLS2 and BAK1 was indeed reduced when GmLMM1 was present. After flg22 stimulation, we observed that GmLMM1 showed a much stronger interaction with NbFLS2 and the NbFLS2-NbBAK1 interaction was suppressed. Thus, GmLMM1 works as a brake to inhibit immune over-activation and controls appropriate immune activation.

Although the exact flg22 receptor and co-receptors remain elusive in soybean, GmLMM1 showed strong interaction with GmFLS2, GmBAK1a, and GmBAK1b, which are the closest homologs of Arabidopsis FLS2 and BAK1/SERKs. Flg22 can trigger the GmFLS2-GmBAK1a and GmFLS2-GmBAK1b interaction, indicating they respond to flg22 and might have the ability to recognize flagellin. In future, it would be interesting to test how soybean rapidly activates PTI pathways by regulating GmLMM1-mediated suppression of PTI during infection by pathogens. We showed that GmLMM1 also strongly suppresses NbRXEG1-mediated immune signaling. Thus, we conclude that GmLMM1 not only suppresses RK-mediated immune signaling, but also suppresses RP-mediated immune signaling. This is a novel report on the malectin-like RK function in RP-mediated signaling. GmLMM1 only slightly suppresses chitin-induced immune responses. We deduced that GmLMM1 might specifically regulate BAK1-dependent immune responses. GmLMM1 possesses autophosphorylation activity, indicating that GmLMM1 is a functional kinase. However, kinase activity is not required for the suppression of PTI responses. Considering that the kinase activity of FER is not required for all functions, it will be interesting to study whether other biological processes are required for GmLMM1's kinase activity. It is also noteworthy that both plant-derived RALF (e.g., RALF23) and pathogen-derived RALF-like peptide may target FER to manipulate plant PTI responses (Masachis *et al*, 2016; Stegmann *et al*, 2017). Identification and characterization of ligands from both plants and pathogens of GmLMM1 will be a subject of future research.

In conclusion, we report the identification of an allelic pair of soybean lesion mimic mutants and characterization of the corresponding gene, *GmLMM1*, encoding a malectin-like RK. The loss-of-function mutant of *GmLMM1* exhibited cell death, PTI activation, and ROS accumulation and confers increased resistance to both soybean bacterial and oomycete pathogens. *GmLMM1* shares close sequence similarity with Arabidopsis *FER* and *ANX1/2*, and its homologs are highly expanded and conserved in the soybean genomes. GmLMM1 constitutively associates with the PRR complexes to negatively regulate PTI and therefore finely tunes PTI levels and balances the fitness cost of immunity responses. Our work illustrates that lesion mimic mutants are useful to study immunity and provides insights into PTI regulation in soybean, which will facilitate future soybean disease resistance breeding through molecular design.

# Materials and Methods

### Plant materials and growth conditions

Soybean cultivars, 'Williams 82", "Hedou 12," and "Dongnong 50" (DN50) obtained from the Chinese Academy of Agricultural Sciences, were used as wild types for this study. *Gmlmm1* mutants (including *Gmlmm1-1* and its allelic mutant *Gmlmm1-2*) were generated from soybean treated with 5% ethyl methane sulfonate. All seedlings of soybean were grown in the field from May to October for separate group construction and gene mapping, or in a climate chamber, where the temperature was maintained at 28°C and humidity at 50% with a 14-h photoperiod for ROS burst experiments and bacterial infection assays. The *Nicotiana benthamiana* plants used for agroinfiltration, microbial

pattern-induced ROS production, subcellular localization, and luciferase complementation assays were cultivated in soil at 23°C and 70% humidity with a 10-h photoperiod. Arabidopsis plants used for protoplast preparation and ROS production assay were grown in soil at 23°C and 70% relative humidity with a 10/14 h day/night cycle.

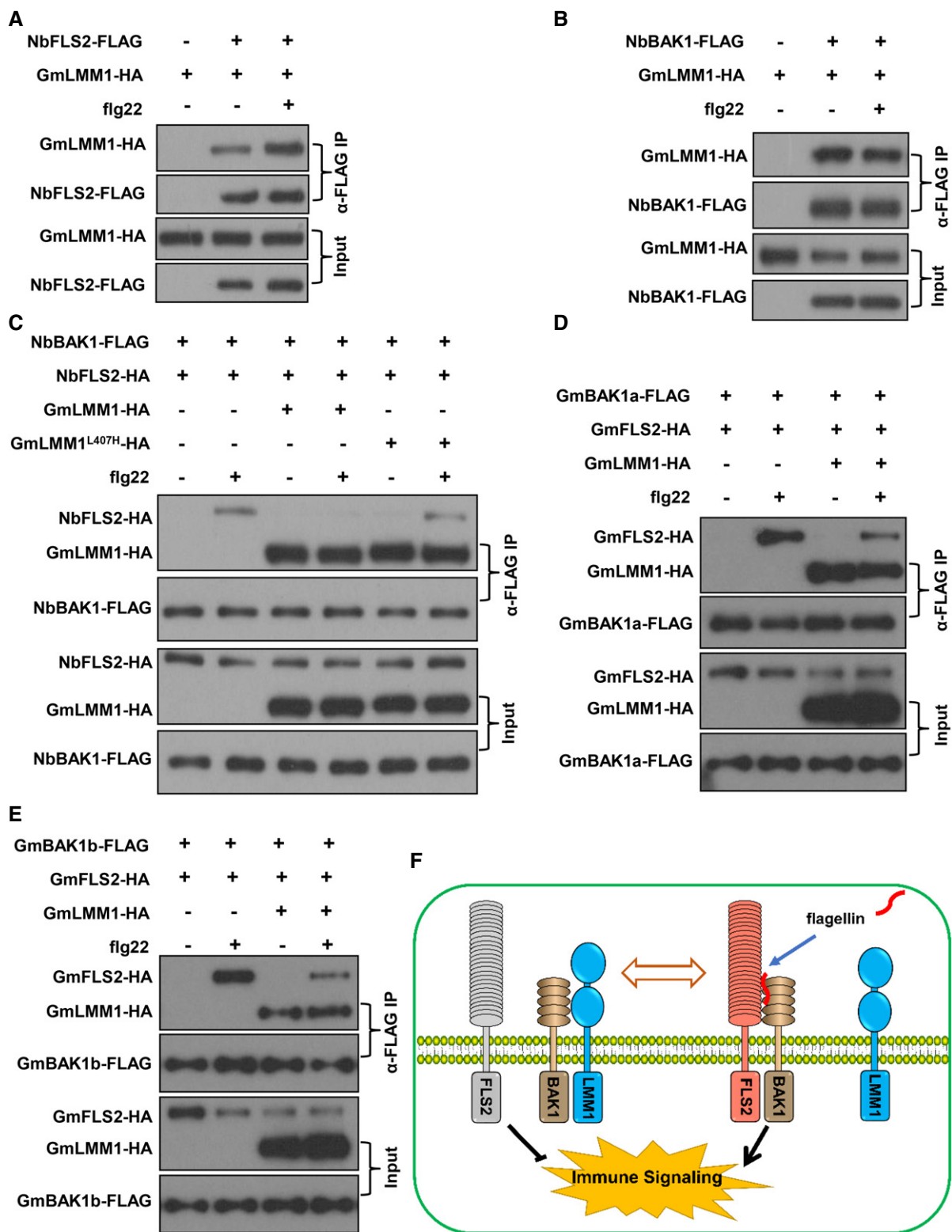

**Figure 6.**

◀

**Figure 6. GMLMM1 interferes with flg22-induced FLS2-BAK1 association.**

A Effect of flg22 on the NbFLS2-GmLMM1 interaction. The indicated constructs were transiently expressed in *N. benthamiana* by *Agrobacterium*-mediated transient expression for 2 days, and the leaves were treated with flg22 (1 μM) or water for 10 min before being subjected to Co-IP assay.

B Regulation of NbBAK1-GmLMM1 interaction by flg22.

C Suppression of flg22-induced NbFLS2-NbBAK1 interaction by GmLMM1. NbFLS2 and NbBAK1 were co-expressed in *N. benthamiana* along with GmLMM1 or GmLMM1$^{L407H}$, treated with H$_2$O or flg22 and subjected to co-IP assays.

D, E Suppression effect of GmLMM1 on the flg22-induced interaction of GmFLS2 with GmBAK1a (D) and GmBAK1b (E).

F A model chart illustrating GmLMM1 mediation of PTI signaling balance in soybean. Flg22 induces the FLS2-BAK1 interaction to trigger downstream immune signaling. GmLMM1 constitutively interacts with BAK1 both before and after immune activation. GmLMM1 can suppress the FLS2-BAK1 interaction to control an appropriate FLS2-BAK1 interaction level. Thus, GmLMM1 works as a molecular switch to regulate moderate immune activation.

Data information: The experiments were performed three times (A–C) or two times (D, E), as biological replicates, with similar results.

Source data are available online for this figure.

## Genetic mapping of *Gmlmm1-1*

In general, the *Gmlmm1-1* mutant was back-crossed for four generations to purify the background and was then further crossed with "Hedou 12" to generate the F2 segregation populations. The F2 populations were subjected to map-based cloning according to methods described previously with the reported molecular markers (Song *et al*, 2015a). According to the experimental re-sequencing data, the INDEL molecular markers were designed and developed about 200-bp upstream and downstream of the heterotopic point of the genomic sequence caused by deletion or insertion between Williams 82 and Hedou 12 for further fine mapping of the *GmLMM1* locus. The fine mapping makers and primers designed in this study are shown in Appendix Table S2 and S7, respectively.

## Plasmid construction and generation of CRISPR lines

To generate the GmLMM1-FLAG construct for transient expression in *Arabidopsis*, the coding sequence was amplified by PCR and cloned into the PUC19-35S-FLAG-RBS vector. For HIS fusion constructs, coding sequences of the *GmLMM1* kinase domain were PCR-amplified and cloned into pET28a. For transient expression in *N. benthamiana* and construction of transgenic Arabidopsis plants, the genomic DNA sequence of *GmLMM1* and coding sequences of other indicated genes were amplified by PCR and cloned into pCAMBIA1300-35S-FLAG/HA-RBS. To check the subcellular localization of GmLMM1 in *N. benthamiana*, the genomic DNA sequence of *GmLMM1* was cloned into pCAMBIA1300-35S-GFP-RBS and introduced into the *Agrobacterium*-mediated transient expression. The K564E mutation for Gm*LMM1* was introduced by site-directed mutagenesis.

To generate the knockout line of Gm*LMM1*, three guide-RNA targeting different regions of *GmLMM1* were designed and cloned into the *VK005-04-soU6-2-GmUbi3* vector (Du *et al*, 2016). The knockout construct was introduced into *Agrobacterium tumefaciens* strain EHA105, which was then transformed to cotyledonary explants of a soybean cultivar DN50 by *Agrobacterium*-mediated transformation (Zhao *et al*, 2016; Li *et al*, 2019).

## Trypan blue and DAB staining

The trypan blue staining assay was implemented as described previously with appropriate modifications (Kachroo *et al*, 2008). Briefly, the samples were boiled for approximately 10 min in trypan blue solution (0.25 mg/ml) and left to stain for 12 h at 25°C.

Subsequently, the samples were de-stained in chloral hydrate (2.5 mg/ml) for approximately 2 days and then analyzed by light microscopy or photographs captured directly with a camera to visualize dead cells. DAB staining was executed as reported previously (Song *et al*, 2015b). In brief, leaves submerged in DAB solution (1 mg/ml, pH 3.8) were stained for 8 h at room temperature in the dark, then transferred to 95% ethanol and boiled for 10 min, followed by incubation in 95% ethanol for several hours to complete chlorophyll removal. Finally, samples were analyzed using light microscopy or photographs captured directly with a camera to detect the accumulation of H$_2$O$_2$.

## Subcellular localization of GmLMM1

The indicated genes were cloned into the pCAMBIA1300-35S-GFP-RBS vector and transiently expressed in *N. benthamiana* leaves by *Agrobacterium*-mediated transient expression. The fluorescence signals were visualized using a Nikon confocal microscope C2 (Japan) under a 488 nm excitation wavelength and a 495–540 nm emission wavelength to monitor the subcellular localizations of GmLMM1 and GmLMM1 $^{L407H}$.

## Bioinformatic analyses

GmLMM1 homologs were obtained by a BLASTP search against Phytozome (version 12.1; http://www.phytozome.net). Multiple sequence alignment was generated using MUSCLE (version 3.8.425) with default parameters. Phylogenetic analyses were performed using neighbor-joining (NJ) methods in MEGA version 7.0. Bootstrap values were performed with 1,000 replications. The protein structure was predicted using the Pfam database (http://pfam.xfam.org/search). A phylogenetic tree was used to determine the closest homologs. Briefly, genes were validated as the closest homologs of *A. thaliana* when their phylogenetic position was in the same clade; identities were calculated based on the alignment of protein sequences.

## Pathogen infection and HR assays

*Phytophthora sojae* infection assays in soybean leaves were performed as previously described with modifications (Song *et al*, 2015b). Briefly, leaves of 12-day-old soybean plants were infected by the *P. sojae* isolate P7076, which was cultured for about 5 days. Subsequently, plants were inoculated for 48 h and 60 h at 25°C in the dark. Data were collected on the infection area, and trypan blue staining was performed to identify soybean resistance to *P. sojae*. To measure

*P. sojae* growth, total DNA was extracted and subjected qPCR analysis using the specific primers. *Psp* and *Psg* infections were performed as previously described (Ashfield *et al*, 1995). In brief, leaves of 10- to 15-day-old soybean plants were infected with bacterium at $5 \times 10^5$ CFU/ml concentration with a needleless syringe. Leaf disks were collected at the indicated time, and the bacterial number was determined 4 days later. For HR assays in *N. benthamiana,* the coding sequences of *Avr3a*, *R3a*, *AvrB,* and *AvrRpt2* were cloned into the pCAMBIA 1300 vector (Bos *et al*, 2009). The constructs were transiently expressed in *N. benthamiana* by *Agrobacterium*-mediated transient expression, and HR was recorded 2 days later. For examination of HR in Arabidopsis, *P. syringae* DC3000 strain containing *AvrB* or *AvrRpt2* was infiltrated into leaves at $OD_{600} = 0.1$ (AvrB) and 0.05 (AvrRpt2), and HR was recorded 3 h (AvrB) and 8 h (AvrRpt2) later.

### ROS production assay

Microbial pattern-triggered ROS burst assays were performed according to a previous report with modifications (Zhang *et al*, 2007). In brief, leaf disks were collected from 20-day-old soil-grown soybean plants or 4- to 5-week-old Arabidopsis plants and were incubated in water for 12 h in a 96-well plate. For ROS measurement in *N. benthamiana* plants, the indicated constructs were transiently expressed in leaves by *Agrobacterium*-mediated transient expression for 2 days, and leaf disks were cut and inoculated in water for 12 h. The samples were treated with luminescence detection buffer [20 mM luminol (Sigma) and 10 mg/ml horseradish peroxidase (Sigma)] containing 1 μM flg22 (Sangon) or 200 μg/ml chitin (Sigma), and the luminescence was recorded using a synergy H1 luminometer.

### Luciferase complementation assay

The split-Luciferase complementation assay was accomplished following a previous description (Zhou *et al*, 2018). Briefly, the indicated Nluc and Cluc constructs were transiently expressed in *N. benthamiana* leaves by *Agrobacterium*-mediated transient expression for 2 days. Leaf disks were cut and incubated with 1 mM luciferin (BioVision) in a 96-well plate for 10–20 min. Luminescence was recorded using a synergy H1 luminometer.

### Co-immunoprecipitation

The indicated constructs were transiently expressed in *N. benthamiana* leaves by *Agrobacterium*-mediated transient expression for 2 days. The leaf samples were collected, ground in liquid nitrogen, and then total protein was extracted using protein IP buffer (50 mM HEPES [pH 7.5], 150 mM KCl, 1 mM EDTA, 0.5% Triton X-100, 1 mM DTT, proteinase inhibitor cocktail). Supernatants were collected by centrifugation at 18,000 *g* for 10 min and incubated with anti-FLAG M2 agarose (Sigma) for 2 h. The agarose was washed four times with IP buffer and eluted with 3 × FLAG peptide (Sigma) for 40 min. The immunoprecipitates were separated on SDS–PAGE gel and detected using the indicated anti-FLAG (Sigma) or anti-HA (CWBIO) immunoblots for detection of flg22-triggered NbFLS2-NbBAK1 interaction. Leaves expressing the indicated constructs were treated with 1 μM flg22 or water for 10 min before the leaf samples were collected.

### *In vitro* phosphorylation assays

The HIS-tagged GmLMM1KD, GmLMM1KD$^{K564E}$, and BIK1 proteins were expressed in *E. coli* and purified using Ni-NTA agarose beads (Qiagen). The purified recombinant proteins were directly subjected to autophosphorylation examination using the pIMAGO-biotin phosphoprotein detection kit (Tymora) according to the manufacturer's instructions.

### qPCR and RNA-seq analysis

Total RNA was extracted from the second trifoliate leaves harvested from Williams 82 and *Gmlmm1-1* at the V2 stage grown in a climate chamber and subjected to RNA extraction using an RNeasy Plant Mini Kit (Qiagen). Total RNA was subjected to qPCR and RNA sequencing analysis. For qPCR analysis, cDNA synthesis was performed using SuperScript III RNA transcriptase (Invitrogen), and the qPCR experiment was performed using the SYBE Premix Ex Taq Kit (TaKaRa).

For transcriptome assay, the leaves of four different plants were mixed to form one sample, and three samples of Williams 82 and *Gmlmm1-1* were collected as three biological repetitions. Paired-end sequencing libraries with an insertion size of about 350 bp were constructed and sequenced on the Illumina Hiseq X Ten platform at Novogene Biotech Company (Beijing, China). Quality control and filtering of off board data to reduce background noise pollution and to remove low-quality reads resulted in more than 40 G of clean bases for the subsequent analysis. Gene expression (RPKM, reads per kilobase per million) levels were estimated using RSEM. Differentially expressed genes (DEGs) were screened with more than a two-fold change $|\log_2 \text{(fold change)}| \geq 1$ and $P < 0.05$ based on the difference in read count between the wild type and mutant using the R package DESeq2. Then, enrichment analysis was carried out using AgriGO, a web-based GO analysis toolkit (http://bioinfo.cau.edu.cn/agriGO/) according to the annotation files of the GO database using the R package cluster Profiler.

### Mass spectrometric analysis of GmLMM1-interacting proteins

To identify GmLMM1-interacting proteins, the coding sequences of *GmLMM1* were amplified by PCR and cloned into the pUC19-35S-FLAG-RBS vector. The GmLMM1-FLAG was transfected into WT (Col-0) Arabidopsis protoplasts and incubated overnight. Total protein was extracted with IP buffer (50 mM HEPES [pH 7.5], 50 mM NaCl, 10 mM EDTA, 0.2% Triton X-100, 0.1 mg/ml Dextran (Sigma), proteinase inhibitor cocktail) and incubated with anti-FLAG M2 agarose (Sigma) for 4 h. The agarose was washed twice with buffer A (50 mM HEPES [pH 7.5], 50 mM NaCl, 10 mM EDTA, 0.1% Triton X-100) and twice with buffer B (50 mM HEPES [pH 7.5], 150 mM NaCl, 10 mM EDTA, 0.1% Triton X-100), and eluted with 3 × FLAG peptide (Sigma) for 40 min. The immunoprecipitates were separated on a 10% SDS–PAGE gel (Invitrogen) and stained with the ProteoSilver Kit (Sigma). The gel was de-stained and digested in-gel with trypsin (10 ng/μl trypsin, 50 mM ammonium bicarbonate [pH 8.0]) at 37°C overnight and then subjected to mass spectrometric analysis by PTM Bio (Hangzhou, China) as previously described (Li *et al*, 2014).

# Data availability

The mass spectrometry proteomics data have been deposited into the ProteomeXchange Consortium via the PRIDE partner repository with the dataset identifier PXD020233 (http://www.ebi.ac.uk/pride/archive/projects/PXD020233; Perez-Riverol *et al*, 2019). Project Name: Identification of GmLMM1-interacting protein in *Arabidopsis* (accession: PXD020233).

Sequences for RNA-seq data were submitted to the National Center for Biotechnology Information (NCBI) with the accession code of BioProject: PRJNA637455 (http://www.ebi.ac.uk/ena/data/view/PRJNA637455) and raw sequencing reads of the transcriptome, SRA: SRP266992 (https://www.ncbi.nlm.nih.gov/sra/?term=SRP266992).

**Expanded View** for this article is available online.

## Acknowledgments
The work was supported by the National Key Research and Development Program of China (2016YFD0101900), and the National Natural Science Foundation of China (31625023).

## Author contributions
DW, XL, YB, SY, XZ, HY, QZ, and GX performed the wet bench and bioinformatical experiments. All authors contributed to experiment design and data analysis. XL, DW, XF and DD wrote the manuscript with final approval of the version to be published by all authors. XF and DD directed the project.

## Conflict of interest
The authors declare that they have no conflict of interest.

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
