## [Review Process File · EMBO Reports]

A Malectin-Like Receptor Kinase Regulates of Cell Death and Pattern-Triggered Immunity in Soybean

Dongmei Wang, Xiangxiu Liang, Yazhou Bao, Suxin Sun, Xiong Zhang, Hui Yu, Qian Zhang, Guangyuan Xu, Xianzhong Feng, and Daolong Dou

DOI: [10.15252/embr.202050442](https://doi.org/10.15252/embr.202050442)

Corresponding author(s): Daolong Dou (ddou@njau.edu.cn) , Xianzhong Feng (fengxianzhong@iga.ac.cn)

Review Timeline:

Submission Date:	16th Mar 20
Editorial Decision:	2nd Apr 20
Revision Received:	3rd Apr 20
Editorial Decision:	23rd Jul 20
Revision Received:	2nd Aug 20
Accepted:	10th Aug 20

Editor: Achim Breiling

Transaction Report:

Dear Prof. Dou,

Thank you for the submission of your research manuscript to EMBO reports. We have now received reports from the three referees that were asked to evaluate your study, which can be found at the end of this email.

As you will see, all referees think that the findings are of interest, but they also have several comments, concerns and suggestions, indicating that a major revision of the manuscript is necessary to allow publication in EMBO reports. As the reports are below, I will not detail them here.

Given the constructive referee comments, we would like to invite you to revise your manuscript with the understanding that all referee concerns must be addressed in the revised manuscript and/or in a detailed point-by-point response. Acceptance of your manuscript will depend on a positive outcome of a second round of review. It is EMBO reports policy to allow a single round of revision only and acceptance of the manuscript will therefore depend on the completeness of your responses included in the next, final version of the manuscript.

Revised manuscripts should be submitted within three months of a request for revision. We are aware that many laboratories cannot function at full efficiency during the current COVID-19/SARS-CoV-2 pandemic and we have therefore extended our 'scooping protection policy' to cover the period required for full revision. Please contact me to discuss the revision should you need additional time, and also if you see a paper with related content published elsewhere.

1) a .docx formatted version of the final manuscript text (including legends for main figures, EV figures and tables), but without the figures included. Please make sure that the changes are highlighted to be clearly visible. Figure legends should be compiled at the end of the manuscript text.

2) individual production quality (high resolution) figure files as .eps, .tif, .jpg (one file per figure), of main figures and EV figures. Please upload these as separate, individual files upon re-submission.

The Expanded View format, which will be displayed in the main HTML of the paper in a collapsible format, has replaced the Supplementary information. You can submit up to 5 images as Expanded View. Please follow the nomenclature Figure EV1, Figure EV2 etc. The figure legend for these should be included in the main manuscript document file in a section called Expanded View Figure Legends after the main Figure Legends section. Additional Supplementary material should be supplied as a single pdf file labeled Appendix. The Appendix should have page numbers and needs

to include a table of content on the first page (with page numbers) and legends for all content. Please follow the nomenclature Appendix Figure Sx, Appendix Table Sx etc. throughout the text, and also label the figures and tables according to this nomenclature.

For more details please refer to our guide to authors:

See also our guide for figure preparation:

http://wol-prod-cdn.literatumonline.com/pb-assets/embosite/EMBOPress_Figure_Guidelines_061115-1561436025777.pdf

4) a complete author checklist, which you can download from our author guidelines (<https://www.embopress.org/page/journal/14693178/authorguide>). Please insert page numbers in the checklist to indicate where the requested information can be found in the manuscript. The completed author checklist will also be part of the RPF.

Please also follow our guidelines for the use of living organisms, and the respective reporting guidelines: <http://www.embopress.org/page/journal/14693178/authorguide#livingorganisms>

5) that primary datasets produced in this study (e.g. RNA-seq, ChIP-seq and array data) are deposited in an appropriate public database. This is now mandatory (like the COI statement). If no primary datasets have been deposited in any database, please state this in this section (e.g. 'No primary datasets have been generated and deposited').

The accession numbers and database should be listed in a formal "Data Availability " section (placed after Materials & Methods) that follows the model below. Please note that the Data Availability Section is restricted to new primary data that are part of this study.

Data availability

6) We strongly encourage the publication of original source data with the aim of making primary data more accessible and transparent to the reader. The source data will be published in a separate source data file online along with the accepted manuscript and will be linked to the relevant figure. If you would like to use this opportunity, please submit the source data (for example scans of entire gels or blots, data points of graphs in an excel sheet, additional images, etc.) of your key experiments together with the revised manuscript. If you want to provide source data, please include size markers for scans of entire gels, label the scans with figure and panel number, and send one PDF file per figure.

8) Regarding data quantification and statistics, can you please specify, where applicable, the number "n" for how many independent experiments (biological replicates) were performed, the bars and error bars (e.g. SEM, SD) and the test used to calculate p-values in the respective figure legends. Please provide statistical testing where applicable, and also add a paragraph detailing this to the methods section. See: <http://www.embopress.org/page/journal/14693178/authorguide#statisticalanalysis>

9) Please provide the abstract written in present tense.

10) Please add scale bars to all microscopic images. Please do not add any writing to them. Please define their size only in the respective figure legend.

11) Please add up to 5 key words to the manuscript text, below the abstract. Please also move the running title there.

12) Please update the format of the references. See: <http://www.embopress.org/page/journal/14693178/authorguide#referencesformat>

Finally, please note that all corresponding authors are required to supply an ORCID ID for their name upon submission of a revised manuscript. Please do that for co-corresponding author Xianzhong Feng. Please find instructions on how to link the ORCID ID to the account in our manuscript tracking system in our Author guidelines:

<http://www.embopress.org/page/journal/14693178/authorguide#authorshipguidelines>

I look forward to seeing a revised version of your manuscript when it is ready. Please let me know if you have questions or comments regarding the revision.

Yours sincerely,

Achim Breiling
Editor
EMBO Reports

Referee #1:

The present manuscript by Wang et al. describes the mapping and characterization of a lesion mimic mutant in soybean, which encodes a malectin domain receptor kinase, similar to Arabidopsis FERONIA. Surprisingly though, the encoding gene (GmLmm1) seems to encode a negative regulator of soybean PTI and cell death control, which is in contrast to the positive regulatory role towards PTI of Arabidopsis FER. GmLMM1 seems to control soybean PTI by association with PRRs at the plasma membrane, a feature GmLMM1 shares with Arabidopsis FER. Overall, the study is well executed and extends our knowledge on PTI in an economically relevant crop species. But I have a few points that should be addressed by the authors before I can recommend publication in EMBO reports.

Specific points:

- The authors state that GmLMM1 does not control R protein function, despite the fact that it was found as the underlying mutation of a lesion mimic mutant. However, the authors conclusion is solely based on the inability of GmLmm1 to suppress R3a-Avr3a triggered HR in *N. benthamiana*. Did the authors test also other R protein pathways? E.g. it was shown for ANX1 to regulate RPS2 and RPM1-based ETI in Arabidopsis. Either one of those could be tested to strengthen the conclusion that GmLMM1 is not regulating ETI.
- gmlmm1 mutants show increased accumulation of defence related genes which is expected as it is an autoimmune mutant. However, the interpretation of the negative regulatory role of GmLMM1 in PTI needs to be interpreted with caution. It is well possible that a true function of the protein is masked by increased autoimmunity, which can increase indirectly PTI responses by elevated steady-state defence hormone accumulation (e.g. SA). Something similar was reported for MPK4 in Arabidopsis, which was long thought to be a negative regulator of PTI, but later found to be actually a positive regulator and the initial conclusions were based on the observation of indirect effects caused by the protein being guarded and the resulting autoimmune phenotype of mpk4 mutants. I would be more convinced if PTI signalling phenotypes were presented from the CRISPR-generated C9-24-13 line, which has a less pronounced lesion mimic phenotype compared to gmlmm1-1 and gmlmm1-2. I am aware that the true nature of GmLmm1 in PTI may be too difficult to unravel in the soybean system. However, a careful discussion of this issue should be incorporated into the manuscript.
- Arabidopsis FERONIA is involved in many physiological responses. Do the soybean GmLmm1 mutants also show phenotypes unrelated to immunity (e.g. fertility, plant morphology, root growth)?
- Otherwise, GmLmm1 phenotype in PTI are more reminiscent of anx1/2 of Arabidopsis. Do the soybean mutants show premature pollen tube rupture?
- Arabidopsis FERONIA and the related ANX1/2 were shown to function as RALF peptide binding receptors. Is GmLMM1 also involved in RALF peptide sensing? The authors could test seedling growth inhibition triggered by RALF1 or a related peptide from soybean and use the various GmLmm1 mutants described in the study

Minor remarks:

- The resolution of some images is quite poor and therefore difficult to judge. In particular, I do not see a clear difference in localization of GmLMM1 and the mutant variant in Fig. 1e. Also, the phylogenetic tree in S2a is of quite poor figure quality.
-

Referee #2:

Through a forward genetic screen, this study identifies mutants of soybean that show lesion-mimic phenotypes and enhanced resistance against bacterial and oomycete pathogens. Positional cloning reveals that these mutants carry mutations in GmLMM1, a malectin-like receptor kinase. CRISPR-Cas9-mediated targeted mutagenesis further support that GmLMM1 is the causal gene for the lesion-mimic phenotype. Using well-designed heterologous experiments, the authors then show that GmLMM1 inhibits PTI but not ETI response, and that GmLMM1 forms complexes with plasma-membrane localized proteins involved in PTI signaling. Co-IP experiments suggest that GmLMM1 inhibits not only formation of Arabidopsis flagellin receptor complex between FLS2 and BAK1, but also complex formation between FLS2 and BAK1 homologues in soybean.

Regulation of plant immunity by malectin-like receptor kinases have been elucidated mostly in the model plant *Arabidopsis thaliana*, while it remains largely unknown whether such regulation occurs in other plant species. The identification and molecular characterization of GmLMM1 in soybean thus broaden our understanding of PTI regulation in plants. I have only several suggestions and comments that may make this nice manuscript even better.

Specific comments:

I think figure 1b and 3c can be combined.

Figure 3c: Here, RNA-seq analysis was performed to identify genes with altered expression in Gmlmm1-1 compared to wild type. A list of genes should be shown with the corresponding log fold change. Also, a more detailed description of RNA-seq and GO-enrichment analysis should be provided for the sake of readers.

Fig. 3e: Lesion size seems to be smaller in the mutants compared to wild type. However, if technically feasible, *P. sojae* growth should be quantified to evaluate resistance against this pathogen. In addition, I see that trypan blue staining was stronger in the mutants than wild type. This finding should be discussed in relation to cell death regulation by GmLMM1.

Supplementary figure 5 and 6: How did they determine the "closest" homologues in soybean based on the phylogenies provided? Please clarify the rationale behind.

Figure 6: These Co-IP experiments nicely showed that GmLMM1 interacts with FLS2 in a ligand-dependent manner and inhibits FLS2-BAK1 complex formation. However, it is still unclear whether inhibition of FLS2-BAK1 complex formation is linked to suppression of PTI by GmLMM1. This point could be addressed by testing whether Gmlmm1-2 protein inhibits FLS2-BAK1 complex formation.

Typos:

Page 3, line 74: ET-TU -> EF-Tu

Page 7, line 179: conservative -> conserved

Referee #3:

The manuscript reports the identification and functional analysis of a malectin-like receptor kinase, GmLMM1, in the regulation of cell death and pattern-triggered immunity in soybean. They have done

a beautiful work by forward genetic screen, map-based cloning and CRISPR-CAS transgenic lines to confirm GmLMM1 is responsible for the lesion mimic phenotype. The Gmlmm1 mutants enhanced soybean resistance against the phytopathogens, including *Pseudomonas syringae* pv. *glycinea* (Psg), *P. sojae* isolate P7076, and GmLMM1 negatively regulated flg22-triggered ROS burst. Mechanistically, GmLMM1 constitutively associated with soybean and tobacco BAK1, and associated with FLS2 upon flg22 stimulation. Moreover, GmLMM1 suppressed flg22-induced FLS2-BAK1 interaction. The manuscript presents some of very nice data on genetic and biochemical analysis of a malectin-like receptor kinase, especially the positional cloning and CRISPR-CAS of GmLMM1 in soybean. Although some malectin-like receptor kinases, such as FER and ANX, have been shown to be involved in plant PTI through association with FLS2 and BAK1, flg22-induced GmLMM1-FLS2 association is unique, and different with FER/ANX-FLS2 constitutive association. Thus, I recommend its publication in EMBO Report with some minor modifications.

Here are some suggestions to improve the manuscript:

1. To support Gmlmm1 are lesion mimic autoimmunity mutants, trypan blue staining or PR expression is suggested.
2. The running title is "Soybean FER regulates cell death and PTI". Do authors have strong evidence that GmLMM1 is a ortholog of FER? FER and ANXs are very close in sequence. The authors also stated in the discussion "Intriguingly, GmLMM1, sharing close sequence and functional similarity to Arabidopsis ANX1 and FER," (line 396). From the function, GmLMM1 is more similar with ANX1 since they are both negative regulators of PTI.
3. The data support the negative role of GmLMM1 in PTI, but did not explain its lesion mimic autoimmunity phenotype. Please discuss this in the discussion.
4. Fig. 1 GmLMM1-GFP is hardly seen.
5. Please label seeding stage in Fig 2a-c, and Fig. 3a and e in the figure legend.
6. The statistical analysis for Fig. 4b.
7. Mistake with the label of the first lanes in Fig. 6a and b (no FLS2/BAK1 in the first lanes). Please label GmFLS2 or GmBAK1 instead of gene number in Fig. 6c-e, and Figure 5.
8. Line 150, "we re-sequenced the mutant ...". Whole genome sequencing or only sequencing GmLMM1 region?
9. The model is hard to understand, and did not convey the message of the manuscript. Please also explain the model in the legend.
10. Some typo, grammar mistakes, or not concise statements. Line 50, "Plants' should be "Plant"; 52, molecule; 74, EF-Tu; line 78, is there any evidence to show CERK1 is a common co-receptor?; 138, p-value-0.07, p>0.05?; 296, interacts; 321, interacts; 339, there is no evidence to show GmLMM1 competes with BAK1; Fig 4a label, Williams82

Referee #1:

The present manuscript by Wang et al. describes the mapping and characterization of a lesion mimic mutant in soybean, which encodes a malectin domain receptor kinase, similar to Arabidopsis FERONIA. Surprisingly though, the encoding gene (GmLmm1) seems to encode a negative regulator of soybean PTI and cell death control, which is in contrast to the positive regulatory role towards PTI of Arabidopsis FER. GmLMM1 seems to control soybean PTI by association with PRRs at the plasma membrane, a feature GmLMM1 shares with Arabidopsis FER. Overall, the study is well executed and extends our knowledge on PTI in an economically relevant crop species. But I have a few points that should be addressed by the authors before I can recommend publication in EMBO reports.

Re: We appreciate your professional comments and suggestions that help us to improve the MS. Please see responses below and the revised MS for detail!

Specific points:

- The authors state that GmLMM1 does not control R protein function, despite the fact that it was found as the underlying mutation of a lesion mimic mutant. However, the authors conclusion is solely based on the inability of GmLmm1 to suppress R3a-Avr3a triggered HR in *N. benthamiana*. Did the authors test also other R protein pathways? E.g. it was shown for ANX1 to regulate RPS2 and RPM1-based ETI in Arabidopsis. Either one of those could be tested to strengthen the conclusion that GmLMM1 is not regulating ETI.

Re: We accepted the suggestions by adding two independent assays. We tested whether AvrB and AvrRpt2-induced HR in *N. benthamiana* by expressing GmLMM1 gene transiently. AvrB or AvrRpt2-induced HR was also examined in the stable GmLMM1-transgenic Arabidopsis lines. As results shown in the newly added Fig EV4 B-C-D, GmLMM1 does not affect both of AvrB and AvrRpt2-induced HR, which is distinct to ANX1. The results and methods were added in the main text accordingly.

- gmLmm1 mutants show increased accumulation of defence related genes which is expected as it is an autoimmune mutant. However, the interpretation of the negative regulatory role of GmLMM1 in PTI needs to be interpreted with caution. It is well possible that a true function of the protein is masked by increased autoimmunity, which can increase indirectly PTI responses by elevated steady-state defence hormone accumulation (e.g. SA). Something similar was reported for MPK4 in Arabidopsis, which was long thought to be a negative regulator of PTI, but later found to be actually a positive regulator and the initial conclusions were based on the observation of indirect effects caused by the protein being guarded and the

resulting autoimmune phenotype of *mpk4* mutants. I would be more convinced if PTI signalling phenotypes were presented from the CRISPR-generated C9-24-13 line, which has a less pronounced lesion mimic phenotype compared to *gmlmm1-1* and *gmlmm1-2*. I am aware that the true nature of GmLmm1 in PTI may be too difficult to unravel in the soybean system. However, a careful discussion of this issue should be incorporated into the manuscript.

Re: We added new results including, (1) the *GmPR1/2* genes were elevated in all the *Gmlmm1-1*, *Gmlmm1-2* and the CRISPR mutants (Fig -EV2), (2) the C9-24-13 line exhibited enhanced ROS accumulation without treatment (Fig EV3A), (3) the line was more resistance to *P. sojae* and *Psg* infection (Fig EV3C-D-E), and (4) the PTI responses (flg22-induced ROS) were also strengthened in this line (Figure 4B). The text was revised accordingly.

We agree that spontaneous cell death of Arabidopsis *mpk4* mutant is caused by that the disruption of MPK4 is guarded by ETI machinery, and the autoimmune phenotypes *mpk4* mutant is indirect. We suggest that GmLMM1 is a negative regulator of plant immunity, which is unlike MPK4. The main reasons include:

(1) As Referee #2 suggested, we expressed GmLMM1^{L407H} in *Nb* plants for protein-interaction assay. We didn't find any cell death phenotypes when it was overexpressed, indicating that GmLMM1^{L407H} cannot be directly guarded at least in *Nb*, but confers cell death phenotype in *Gmlmm1-2*.

(2) GmLMM1 seems more preferentially to regulate BAK1-dependent PTI responses. This specificity and selectivity in PTI inhibition (flg22/XEG1, but much weaker for chitin and not for ETI) suggest that it regulates PTI directly.

(3) We also provided several lines of evidence to prove GmLMM1 regulates PTI directly: PTI in loss-of-mutant, stable transgenic Arabidopsis lines, transient expression in *Nb*, and biochemical association with PRR complex.

Of course, we could not exclude the possibility, by which an unknown NLR protein may guard GmLMM1-mediated disruption MAPK kinase cascades in soybean. The discussion was revised accordingly.

- Arabidopsis FERONIA is involved in many physiological responses. Do the soybean GmLmm1 mutants also show phenotypes unrelated to immunity (e.g. fertility, plant morphology, root growth)?**
- Otherwise, GmLmm1 phenotype in PTI is more reminiscent of *anx1/2* of Arabidopsis. Do the soybean mutants show premature pollen tube rupture?**

Re: We combine these two questions together to response.

As shown in Appendix Table S5, some agronomic traits, including plant height, seed numbers of per pod, were significantly altered in one mutant (*Gmlmm1-2* as an example). The detail results were added in the main text accordingly.

Actually, we are carrying out an independent project on regulation of GmLMM1 on physiological responses, pollen tube development, and reproduction. Some of our unpublished data below showed that *GmLMM1* gene is also highly expressed in flower organs, and both mutants exhibited defects on pollen development. Thus, we suppose that this gene has wide functions in some other processes. By the way, we are also trying to complement *GmLMM1* gene in Arabidopsis mutants of *fer*, *anx1* and *anx1 anx2*, but failed many times.

Unpublished data removed at the authors' request

- Arabidopsis FERONIA and the related ANX1/2 were shown to function as RALF peptide binding receptors. Is GmLMM1 also involved in RALF peptide sensing? The authors could test seedling growth inhibition triggered by RALF1 or a related peptide from soybean and use the various GmLmm1 mutants described in the study.

Re: A good suggestion! We identified soybean RALF peptides bioinformatically and found >50 members. It will take 1-2 years to figure out underlying mechanisms adapted by GmLMM1 to sense its corresponding RALF. We hope the seedling growth inhibition assay will accelerate the process.

Currently, we used AtRALF1, a well-studied peptide, to perform the root growth inhibition assay. As the results below shown, we are interested to find that AtRALF1 can inhibit root growth, but the inhibition is not related to GmLMM1. We speculate that AtRALF1 may be recognized by another unknown soybean gene and the RALF corresponding to GmLMM1 need discovery. We didn't add the results in the current MS because it is not highly related.

Unpublished data removed at the authors' request

Minor remarks:

- The resolution of some images is quite poor and therefore difficult to judge. In particular, I do not see a clear difference in localization of GmLMM1 and the mutant variant in Fig. 1e. Also, the phylogenetic tree in S2a is of quite poor figure quality.

Re: The low resolution of the figures is caused by that we submitted all the figures together with figure legends in a single PDF file. Here, all figures were carefully prepared following the instructions and submitted in a .tif file separately.

Referee #2:

Through a forward genetic screen, this study identifies mutants of soybean that show lesion-mimic phenotypes and enhanced resistance against bacterial and oomycete pathogens. Positional cloning reveals that these mutants carry mutations in GmLMM1, a malectin-like receptor kinase. CRISPR-Cas9-mediated targeted mutagenesis further support that GmLMM1 is the causal gene for the lesion-mimic phenotype. Using well-designed heterologous experiments, the authors then show that GmLMM1 inhibits PTI but not ETI response, and that GmLMM1 forms complexes with plasma-membrane localized proteins involved in PTI signaling. Co-IP experiments suggest that GmLMM1 inhibits not only formation of Arabidopsis flagellin receptor complex between FLS2 and BAK1, but also complex formation between FLS2 and BAK1 homologues in soybean.

Regulation of plant immunity by malectin-like receptor kinases have been elucidated mostly in the model plant *Arabidopsis thaliana*, while it remains largely unknown whether such regulation occurs in other plant species. The identification and molecular characterization of GmLMM1 in soybean thus broaden our understanding of PTI regulation in plants. I have only several suggestions and comments that may make this nice manuscript even better.

Re: We appreciate your professional comments and suggestions that help us to improve the MS.

Specific comments:

I think figure 1b and 3c can be combined.

Re: We guess that you mean combine Figure 3d (not 3c) with 1b together because figure 1b and 3d are similar experiments. We accepted this.

Figure 3c: Here, RNA-seq analysis was performed to identify genes with altered expression in Gmlmm1-1 compared to wild type. A list of genes should be shown with the corresponding log fold change. Also, a more detailed description of RNA-seq and GO-enrichment analysis should be provided for the sake of readers.

Re: We added one file as the Source data-RNA-seq including the genes with the altered expression and the Go -enrichment analysis.

Fig. 3e: Lesion size seems to be smaller in the mutants compared to wild type. However, if technically feasible, *P. sojae* growth should be quantified to evaluate resistance against this pathogen. In addition, I see that trypan blue staining was

stronger in the mutants than wild type. This finding should be discussed in relation to cell death regulation by GmLMM1.

Re: We accepted this suggestion by quantifying *P. sojae* growth with qPCR analysis. The results were added as Fig EV3A and the text were revised accordingly.

The relation to cell death regulation by GmLMM1 was discussed in the text.

Supplementary figure 5 and 6: How did they determine the "closest" homologues in soybean based on the phylogenies provided? Please clarify the rationale behind.

Re: Briefly, the "closest" homologues were determined by blast analysis, phylogenetic tree and identity calculation. The section of Methods was modified accordingly. By the way, the result was revised to:

“The closest homologs of GmLMM1 in Arabidopsis are the malectin-like RKs, FER (identity = 50.5%) and ANX1 (identity = 43.8%) (Fig 1D and Fig EV1E).”

Figure 6: These Co-IP experiments nicely showed that GmLMM1 interacts with FLS2 in a ligand-dependent manner and inhibits FLS2-BAK1 complex formation. However, it is still unclear whether inhibition of FLS2-BAK1 complex formation is linked to suppression of PTI by GmLMM1. This point could be addressed by testing whether Gmlmm1-2 protein inhibits FLS2-BAK1 complex formation.

Re: A good suggestion. We added GmLMM1^{L407H} for testing and indeed found that the mutation in *Gmlmm1-2* lost its ability to suppress NbFLS2-NbBAK1 interaction. Figure 6C and the text were revised accordingly.

Typos:

Page 3, line 74: ET-TU -> EF-Tu

Page 7, line 179: conservative -> conserved

Re: Corrected.

Referee #3:

The manuscript reports the identification and functional analysis of a malectin-like receptor kinase, GmLMM1, in the regulation of cell death and pattern-triggered immunity in soybean. They have done a beautiful work by forward genetic screen, map-based cloning and CRISPR-CAS transgenic lines to confirm GmLMM1 is responsible for the lesion mimic phenotype. The *Gmlmm1* mutants enhanced soybean resistance against the phytopathogens, including *Pseudomonas syringae* pv. *glycinea* (Psg), *P. sojae* isolate P7076, and GmLMM1 negatively regulated flg22-triggered ROS burst. Mechanistically, GmLMM1 constitutively associated with soybean and tobacco BAK1, and associated with FLS2 upon flg22 stimulation. Moreover, GmLMM1 suppressed flg22-induced FLS2-BAK1 interaction. The manuscript presents some of very nice data on genetic and biochemical analysis of a malectin-like receptor kinase, especially the positional cloning and CRISPR-CAS of GmLMM1 in soybean. Although some malectin-like receptor kinases, such as FER and ANX, have been shown to be involved in plant PTI through association with FLS2 and BAK1, flg22-induced GmLMM1-FLS2 association is unique, and different with FER/ANX-FLS2 constitutive association. Thus, I recommend its publication in EMBO Report with some minor modifications.

Re: We appreciate your professional comments and suggestions that help us to improve the MS.

Here are some suggestions to improve the manuscript:

1. To support *Gmlmm1* are lesion mimic autoimmunity mutants, trypan blue staining or PR expression is suggested.

Re: We accepted this good suggestion by performing the trypan blue staining and *GmPR1/2* genes expression assays in *Gmlmm1-1/2* and a CRISPR line C9-24-13. As shown in the newly added Fig EV2, the mutant lines showed more cell death and enhanced PR genes expression the WT controls. The text was revised accordingly.

2. The running title is "Soybean FER regulates cell death and PTI". Do authors have strong evidence that GmLMM1 is a ortholog of FER? FER and ANXs are very close in sequence. The authors also stated in the discussion "Intriguingly, GmLMM1, sharing close sequence and functional similarity to Arabidopsis ANX1 and FER," (line 396). From the function, GmLMM1 is more similar with ANX1 since they are both negative regulators of PTI.

Re: We agreed that “GmLMM1 is more similar with ANX1 since they are both negative regulators of PTI”. The running title has changed to ‘Soybean RK regulates cell death and PTI’. Besides, we added “The closest homologs of GmLMM1 in Arabidopsis are the malectin-like RKs, FER (identity = 50.5%) and ANX1 (identity = 43.8%)” to clarify the sequence similarity between GmLMM1 and FER/ANX1.

3. The data support the negative role of GmLMM1 in PTI, but did not explain its lesion mimic autoimmunity phenotype. Please discuss this in the discussion.

Re: We adding two parts of explanation labeled with red face in discussion section.

4. Fig. 1 GmLMM1-GFP is hardly seen.

Re: We have repeated the experiment and changed a new graph.

5. Please label seeding stage in Fig 2a-c, and Fig. 3a and e in the figure legend.

6. The statistical analysis for Fig. 4b.

7. Mistake with the label of the first lanes in Fig. 6a and b (no FLS2/BAK1 in the first lanes). Please label GmFLS2 or GmBAK1 instead of gene number in Fig. 6c-e, and Figure 5.

Re: Accepted all.

8. Line 150, "we re-sequenced the mutant ..." Whole genome sequencing or only sequencing GmLMM1 region?

Re: It was changed to “we performed the whole genome sequencing of the mutant *Gmlmm1-1* and found...”.

9. The model is hard to understand, and did not convey the message of the manuscript. Please also explain the model in the legend.

Re: We have made some modifications on the model chart and explain the model in the legend.

10. Some typo, grammar mistakes, or not concise statements. Line 50, "Plants' should be "Plant"; 52, molecule; 74, EF-Tu; line 78, is there any evidence to show CERK1 is a common co-receptor?; 138, p-vale-0.07, $p > 0.05$?; 296, interacts; 321, interacts; 339, there is no evidence to show GmLMM1 competes with BAK1; Fig 4a label, Williams82

Re: Corrected the typos thoroughly.

Dear Prof. Dou,

Thank you for the submission of your revised manuscript to our editorial offices. I have now received the reports from the referees that were asked to re-evaluate your study, you will find below. As you will see, the referees now support the publication of your study in EMBO reports, but have some final comments/further suggestions to improve the manuscript, we ask you to address in a final revised version of the manuscript. Please also provide a point-by-point-response addressing the remaining points by the referees.

Further, I have these editorial requests:

- I suggest this slightly changed title (added 'a'):

Regulation of Cell Death and Pattern-Triggered Immunity by a Malectin-Like Receptor Kinase in Soybean

- Please use for the final manuscript text our new reference format:

- As indicated by the referees, please have your manuscript carefully proofread by a native speaker.

- You indicate in the figure legends several times that n represents technical repeats. It seems though, that these are biological replicates. E.g. in Fig. 1 you state that 'Leaves from 2-week-old soybean plants were infected with Psg and the bacteria number was determined at 0 and 4 dpi.' If here each time a different plant was used, aren't these then biological replicates? Please check, also in the other legends, or explain.

- Could statistical testing also be performed for the data shown in Fig. 5B/E and EV5 C/D?

- Please provide also the Western blot source data for figures 4 and EV4. Please check that source data for all Western blots in the manuscript are provided.

- Please upload the source data one file per figure. Thus, please provide separated excel tables for each figure. If there is more than one file per figure, please ZIP these together before uploading.

- Please remove the Appendix table information from the end of the main text. This is now included in the Appendix itself.

- You indicate in the author checklist that your study could fall under dual use research restriction. Please detail this in response to this message, and in the author checklist. Dual Use Research of Concern (DURC) is life sciences research that, based on current understanding, can be reasonably anticipated to provide knowledge, information, products, or technologies that could be directly misapplied to pose a significant threat with broad potential consequences to public health and safety, agricultural crops and other plants, animals, the environment, materiel, or national security. If this is really the case here, we need a detailed explanation to decide if we proceed with publication. See also:

<http://www.embopress.org/page/journal/14693178/authorguide#biosecurity>

- Finally, please find attached a word file of the manuscript text (provided by our publisher) with

changes we ask you to include in your final manuscript text, and some queries, we ask you to address. It seems these points have already been addressed, but please check again. Please provide your final manuscript file with track changes, in order that we can see any modifications done.

In addition I would need from you:

- a short, two-sentence summary of the manuscript
- three bullet points highlighting the key findings of your study
- a schematic summary figure (in jpeg or tiff format with the exact width of 550 pixels and a height of not more than 400 pixels) that can be used as a visual synopsis on our website.

Kind regards,

Achim Breiling
Editor
EMBO Reports

Referee #1:

The authors adequately responded to the previously raised criticism and provide sufficient new data to make the manuscript sufficient for acceptance in EMBO reports. Overall a very nice report on soybean malectin-RKs with novel findings, also in the context of Arabidopsis research, as a function of a malectin-RK in negative regulation of cell death responses is not known to date.

I have only a few minor remarks that the authors should address:

- There was a misunderstanding in my previous comments regarding potential guarding of GmLMM1 in soybean, which may be similar to Arabidopsis mpk4 mutants. My intention was to highlight, that the increased PTI responses in soybean mutants may be an indirect effect caused by the increased steady-state SA accumulation (as shown by the authors through increased PR1 expression in gmlmm1 mutants). However, there are more lines of evidence in addition to the loss of function soybean phenotype that indeed suggest that GmLMM1 is a negative regulator of PTI (Transgenic Arabidopsis lines overexpressing GmLMM1 and resulting in reduced flg22-triggered ROS; transient expression in N. benthamiana with reduced ROS). Therefore, this discussion point should be removed or modified, stating that the increased PTI in gmlmm1 could be explained by increased steady state PR1/SA, but additional data support the notion that it is a negative regulator of PTI
- Careful proof reading should be done, as there are still some typos and little English grammar mistakes in the manuscript

Referee #2:

This manuscript has been greatly improved with the addition of new experimental data that answer the concerns raised by the reviewers and that further support the role of GmLMM1 in PTI regulation. The findings presented here would be a great contribution to the field.

Please consider the following minor comment.

Line 409-418: Here, the mechanism of spontaneous cell death caused by gmlmm1 mutation is discussed. However, this discussion seems to be too speculative and some parts are not convincing. For example, what made you believe that "the spontaneous cell death of Gmlmm1 could not be fully explained by its interaction with PRRs"? This point is not addressed in the present manuscript. In addition, as BAK1 is a member of SERK family proteins, it would be more appropriate to state "cell death-related components, e.g. BAK1 and other SERK family proteins".

There are some typos and errors.

Line 36: to controls -> to control.

Line 168: genotypes their -> genotypes of their

Line 222: cell death. We -> cell death, we

The order of fig 4 and 5 is inverted.

Line 278: The sentence regarding AvrB and AvrRpt2-triggered cell death in *N. benthamiana* requires a reference(s).

Line 374: delete "using".

Line 450: PRR complexes formation -> PRR complex formation

Referee #3:

The revision is satisfactory. A small suggestion: Change the running title 'Soybean RK regulates cell death and PTI' to 'Soybean CrRLK1L gene GmLMM1 regulates cell death and PTI'.

Dear Prof. Dou,

Thank you for the submission of your revised manuscript to our editorial offices. I have now received the reports from the referees that were asked to re-evaluate your study, you will find below. As you will see, the referees now support the publication of your study in EMBO reports, but have some final comments/further suggestions to improve the manuscript, we ask you to address in a final revised version of the manuscript. Please also provide a point-by-point-response addressing the remaining points by the referees.

Further, I have these editorial requests:

- I suggest this slightly changed title (added 'a'):
Regulation of Cell Death and Pattern-Triggered Immunity by a Malectin-Like Receptor Kinase in Soybean.

R : we change the title to 'A Malectin-Like Receptor Kinase Regulates of Cell Death and Pattern-Triggered Immunity in Soybean' because your suggestion will make 1 character over as the system indicates.

- Please use for the final manuscript text our new reference format:
<http://www.embopress.org/page/journal/14693178/authorguide#referencesformat>

R : We have reorganized the reference and citation according to the guidelines.

- As indicated by the referees, please have your manuscript carefully proofread by a native speaker.

R : Our MS is now edited by a professional language editing company, Letpub. Please see the following photo for certification!

Manuscript Title:
Regulation of Cell Death and Pattern-Triggered Immunity by a Malectin-Like
Receptor Kinase in Soybean

Date of Revision
July 30, 2020

Abstract:
Plant cells can sense conserved molecular patterns through cell surface-localized pattern-recognition receptors (PRRs) and can initiate pattern-triggered immunity (PTI). Details of the PTI signaling network are starting to be uncovered in Arabidopsis but are still poorly understood in other species, including soybean (*Glycine max*). In this study, we performed a forward genetic screen for autoimmunity-related lesion mimic mutants (lmms) in soybean and identified two allelic mutants, which carry mutations in *Glyma.19G054400*, encoding a malectin-like receptor kinase (RLK). The mutants exhibited enhanced resistance to both bacterial and oomycete pathogens, as well as elevated ROS production upon treatment with the bacterial pattern flag22. Overexpression of the *GmLMM1* gene in *Nicotiana benthamiana* severely suppressed flag22-triggered ROS production and oomycete pattern XEG1-induced cell death. We further showed that *GmLMM1* interacted...

This document certifies that the manuscript listed above was copy edited for proper English language at LetPub. All of our language editors are native English speakers with long-term experience in editing scientific and technical manuscripts. We are committed to leveling the playing field for researchers whose native language is not English.

- Neither the research content nor the authors' intended meaning were altered in any way during the editing process.
- Documents receiving this certification should be considered ready for publication where language issues are concerned.
- However, the authors may accept or reject LetPub's suggestions and changes at their own discretion.
- If you have any questions or concerns about this edited document, please contact us at support@letpub.com.

LetPub is an author service brand owned and operated by Horizon LLC, headquartered in the Boston area, we are a full-featured author services company with a large team of US-based certified language and scientific editors, ISO 17020 accredited translators, and professional scientific illustrators and animators. We advocate ethical publication practices and are an official member of the Committee on Publication Ethics (COPE).

For more information about our company, services, and partnership programs, please visit www.letpub.com.
© 2020 Horizon, LLC. All Rights Reserved. Tel: +1-151-422-9698 Email: info@horizon.com Address: 400 Pitts Ave, Suite 502, Watertown, MA 02451, United States

- You indicate in the figure legends several times that n represents technical repeats. It seems though, that these are biological replicates. E.g. in Fig. 1 you state that 'Leaves from 2-week-old soybean plants were infected with Psg and the bacteria number was determined at 0 and 4 dpi.' If here each time a different plant was used, aren't these then biological replicates? Please check, also in the other legends, or explain.

R : We are sorry for the inappropriate description. The n value represents the number of samples used in each experiment and it cannot be considered as technical repeats. We have corrected this in the figure legends.

Strictly speaking, biological repeat means completely independent experiments performed at different time. We have already listed the biological repeat number at the bottom of figure legends. Thus, we have redefined the n value as "sample number".

- Could statistical testing also be performed for the data shown in Fig. 5B/E and EV5 C/D?

R : We have added statistical analysis for Fig. 5B/E and EV5 C/D.

- Please provide also the Western blot source data for figures 4 and EV4. Please check that source data for all Western blots in the manuscript are provided.

R : Now, original data for all the western-blot are provided in the source data.

- Please upload the source data one file per figure. Thus, please provide separated excel tables for each figure. If there is more than one file per figure, please ZIP these together before uploading.

R : As requested, we provided one source data file for each figure.

- Please remove the Appendix table information from the end of the main text. This is now included in the Appendix itself.

R : Accepted.

- You indicate in the author checklist that your study could fall under dual use research restriction. Please detail this in response to this message, and in the author checklist. Dual Use Research of Concern (DURC) is life sciences research that, based on current understanding, can be reasonably anticipated to provide knowledge, information, products, or technologies that could be directly misapplied to pose a significant threat with broad potential consequences to public health and safety, agricultural crops and other plants, animals, the environment, materiel, or national security. If this is really the case here, we need a detailed explanation to decide if we proceed with publication. See also: <http://www.embopress.org/page/journal/14693178/authorguide#biosecurity>

R : in the author checklist, we indicated that 'our study can fall under dual use research restriction.' because:

Our study falls to fundamental research in plant science area and could not provide knowledge, information, products, or technologies that could be applied directly. For example, the plant pathogens are all model isolates and have been widely used in the labs; none of them are harmful to the plants in the fields. The transgenic plant lines that were generated in this study are all carefully managed under strict national policy. Thus, in our opinions, the study is not related to dual use research of concern.

Here, we update it to 'Not Applicable. The study is not related to dual use research of concern.'

Please let us know if it is not correct! Many thanks!

- Finally, please find attached a word file of the manuscript text (provided by our publisher) with changes we ask you to include in your final manuscript text, and some queries, we ask you to address. It seems these points have already been addressed, but please check again. Please provide your final manuscript file with track changes, in order that we can see any modifications done.

R : We have corrected. The changes were labeled with red face in main text.

In addition I would need from you:

- a short, two-sentence summary of the manuscript
- three bullet points highlighting the key findings of your study

Summary

We identify a soybean malectin-like RK gene, *GmLMM1*, from an allelic pair of soybean lesion mimic mutants, and demonstrate that it may associates with PRR complexes to negatively regulate PTI responses.

Three bullet points highlighting the key findings

- 1. A forward genetic screening for lesion mimic mutants identified a malectin-like RK *GmLMM1*.**
- 2. *GmLMM1* is involved in the PRR complex and works as a negative regulator of plant immunity.**
- 3. *GmLMM1* regulates the interaction between *FLS2* and *BAK1* to control plant immune activation.**

- a schematic summary figure (in jpeg or tiff format with the exact width of 550 pixels and a height of not more than 400 pixels) that can be used as a visual synopsis on our website.

R : Accepted.

R : Thanks for your great effort in handling our MS and valuable suggestions!

Kind regards,

Achim Breiling
Editor
EMBO Reports

Referee #1:

The authors adequately responded to the previously raised criticism and provide sufficient new data to make the manuscript sufficient for acceptance in EMBO reports. Overall a very nice report on soybean malectin-RKs with novel findings,

also in the context of Arabidopsis research, as a function of a malectin-RK in negative regulation of cell death responses is not known to date.

I have only a few minor remarks that the authors should address:

- There was a misunderstanding in my previous comments regarding potential guarding of GmLMM1 in soybean, which may be similar to Arabidopsis mpk4 mutants. My intention was to highlight, that the increased PTI responses in soybean mutants may be an indirect effect caused by the increased steady-state SA accumulation (as shown by the authors through increased PR1 expression in gmlmm1 mutants). However, there are more lines of evidence in addition to the loss of function soybean phenotype that indeed suggest that GmLMM1 is a negative regulator of PTI (Transgenic Arabidopsis lines overexpressing GmLMM1 and resulting in reduced flg22-triggered ROS; transient expression in *N. benthamiana* with reduced ROS). Therefore, **this discussion point should be removed or modified**, stating that the increased PTI in gmlmm1 could be explained by increased steady state PR1/SA, but additional data support the notion that it is a negative regulator of PTI.

R : We agree you points and have removed this part of discussion.

- Careful proof reading should be done, as there are still some typos and little English grammar mistakes in the manuscript.

R : Our MS is now edited by a native speaker from Letpub company.

Referee #2:

This manuscript has been greatly improved with the addition of new experimental data that answer the concerns raised by the reviewers and that further support the role of GmLMM1 in PTI regulation. The findings presented here would be a great contribution to the field.

Please consider the following minor comment.

Line 409-418: Here, the mechanism of spontaneous cell death caused by gmlmm1 mutation is discussed. However, this discussion seems to be too speculative and some parts are not convincing. For example, what made you believe that "**the spontaneous cell death of Gmlmm1 could not be fully explained by its interaction with PRRs**"? This point is not addressed in the present manuscript. In addition, as BAK1 is a member of SERK family proteins, it would be more appropriate to state "cell death-related components, e.g. BAK1 and other SERK family proteins".

R : We have made some corrections in the discussion.

There are some typos and errors.

Line 36: to controls -> to control.

Line 168: genotypes their -> genotypes of their

Line 222: cell death. We -> cell death, we

The order of fig 4 and 5 is inverted.

Line 278: The sentence regarding AvrB and AvrRpt2-triggered cell death in *N. benthamiana* requires a reference(s).

Line 374: delete "using".

Line 450: PRR complexes formation -> PRR complex formation

R : All are corrected.

Referee #3:

The revision is satisfactory. A small suggestion: Change the running title 'Soybean RK regulates cell death and PTI' to 'Soybean CrRLK1L gene GmLMM1 regulates cell death and PTI'.

R : The running title should less than 40 characters including spaces.

Thus, we keep the original writing.

Again, we appreciate the editors and reviewers for your professional and helpful suggestions on our manuscript.

Prof. Daolong Dou
China Agricultural Univeristy
#2 Yuanmingyuan West Rd.
Beijing, Beijing 100193
China

Dear Prof. Dou,

I am very pleased to accept your manuscript for publication in the next available issue of EMBO reports. Thank you for your contribution to our journal.

At the end of this email I include important information about how to proceed. Please ensure that you take the time to read the information and complete and return the necessary forms to allow us to publish your manuscript as quickly as possible.

As part of the EMBO publication's Transparent Editorial Process, EMBO reports publishes online a Review Process File to accompany accepted manuscripts. As you are aware, this File will be published in conjunction with your paper and will include the referee reports, your point-by-point response and all pertinent correspondence relating to the manuscript.

If you do NOT want this File to be published, please inform the editorial office within 2 days, if you have not done so already, otherwise the File will be published by default [contact: emboreports@embo.org]. If you do opt out, the Review Process File link will point to the following statement: "No Review Process File is available with this article, as the authors have chosen not to make the review process public in this case."

Should you be planning a Press Release on your article, please get in contact with emboreports@wiley.com as early as possible, in order to coordinate publication and release dates.

Thank you again for your contribution to EMBO reports and congratulations on a successful publication. Please consider us again in the future for your most exciting work.

Yours sincerely,

Achim Breiling
Editor
EMBO Reports

THINGS TO DO NOW:

You will receive proofs by e-mail approximately 2-3 weeks after all relevant files have been sent to our Production Office; you should return your corrections within 2 days of receiving the proofs.

Please inform us if there is likely to be any difficulty in reaching you at the above address at that time. Failure to meet our deadlines may result in a delay of publication, or publication without your corrections.

All further communications concerning your paper should quote reference number EMBOR-2020-50442V3 and be addressed to emboreports@wiley.com.

Should you be planning a Press Release on your article, please get in contact with emboreports@wiley.com as early as possible, in order to coordinate publication and release dates.

Corresponding Author Name: Daolong Dou

Manuscript Number: EMBOR-2020-50442V2